# Learning Neural Causal Models with Active Interventions

## Abstract

Discovering causal structures from data is a challenging inference problem of fundamental importance in all areas of science. The appealing scaling properties of neural networks have recently led to a surge of interest in differentiable neural network-based methods for learning causal structures from data. So far, differentiable causal discovery has focused on static datasets of observational or interventional origin. In this work, we introduce an active intervention-targeting mechanism which enables quick identification of the underlying causal structure of the data-generating process. Our method significantly reduces the required number of interactions compared with random intervention targeting and is applicable for both discrete and continuous optimization formulations of learning the underlying directed acyclic graph (DAG) from data. We examine the proposed method across multiple frameworks in a wide range of settings and demonstrate superior performance on multiple benchmarks from simulated to real-world data.

## 1 Introduction

Learning causal structure from data is a challenging but important task that lies at the heart of scientific reasoning and accompanying progress in many disciplines (Sachs et al., 2005; Hill et al., 2016; Lauritzen & Spiegelhalter, 1988; Korb & Nicholson, 2010). While there exists a plethora of methods for the task, computationally and statistically more efficient algorithms are highly desirable (Heinze-Deml et al., 2018). As a result, there has been a surge in interest in differentiable structure learning and the combination of deep learning and causal inference (Schölkopf et al., 2021). Such methods define a structural causal model with smoothly differentiable parameters that are adjusted to fit observational data (Zheng et al., 2018; Yu et al., 2019; Zheng et al., 2020; Bengio et al., 2019; Lorch et al., 2021; Annadani et al., 2021), although some methods can accept interventional data, thereby significantly improving the identification of the underlying data-generating process (Ke et al., 2019; Brouillard et al., 2020; Lippe et al., 2021). However, the improvement critically depends on the experiments and interventions available to the learner.

Despite advances in high-throughput methods for interventional data in specific fields (Dixit et al., 2016), the acquisition of interventional samples in general settings tends to be costly, technically impossible or even unethical for specific interventions. There is, therefore, a need for efficient usage of the available interventional samples and efficient experimental design to keep the number of interventions to a minimum.

A significant amount of prior work exists in causal structure learning that leverages active learning and experimental design to improve identifiability in a sequential manner. These approaches are either graph theoretical (He & Geng, 2008; Eberhardt, 2012; Hyttinen et al., 2013; Hauser & Bühlmann, 2014; Shanmugam et al., 2015; Kocaoglu et al., 2017b;a; Lindgren et al., 2018; Ghassami et al., 2018; 2019; Greenewald et al., 2019; Squires et al., 2020), Bayesian (Murphy, 2001; Tong & Koller, 2001; Masegosa & Moral, 2013; Cho et al., 2016; Ness et al., 2017; Agrawal et al., 2019; Zemplenyi & Miller, 2021) or rely on Invariant Causal Prediction (Gamella & Heinze-Deml, 2020). These methods are typically computationally very expensive and do not scale well with respect to the number of variables or dataset size (Heinze-Deml et al., 2018). A promising alternative is the use of active learning in a continuous optimization framework for causal structure learning from joint data. However, since the applicability of existing scores / heuristics for selecting inter-

vention targets is limited for existing frameworks (see §A.1), current approaches rely on random and independent interventions and do not leverage the acquired evidence from processed experiments.

We thus propose a novel method of active selection of intervention targets that can easily be incorporated into many differentiable causal discovery algorithms. Since most of these algorithms treat the adjacency matrix of the causal graph as a learned soft-adjacency, it is readily available for parametrized sampling of different hypothesis graphs. Our method looks for an intervention target that gives maximum disagreement between post-interventional sample distributions under these hypothesis graphs. We conjecture that interventions on such nodes will contain more information about the causal structure and hence enable more efficient learning. To the best of our knowledge, our paper is the first approach to combine both a continuous optimization framework and *active causal structure learning* from observational and interventional data. We summarize our contributions as follows:

- We propose a novel approach for selecting interventions (single and multi-target) which identify the underlying graph efficiently and can be used for any differentiable causal discovery method.
- We introduce a novel, scalable two-phase DAG sampling procedure which efficiently generates hypothesis DAGs based on a soft-adjacency matrix.
- We examine the proposed intervention-targeting method across multiple differentiable causal discovery frameworks in a wide range of settings and demonstrate superior performance against established competitive baselines on multiple benchmarks from simulated to real-world data.
- We provide empirical insights on the distribution of selected intervention targets and its connection to the (causal) topological order of the variables in the underlying system.

## 2 PRELIMINARIES

**Structural Causal Model.** An SCM (Peters et al., 2017) is defined over a set of random variables $X_1, \ldots, X_M$ or just $X$ for short and a directed acyclic graph (DAG) $G = (V, E)$ over variable nodes $V = \{1, \ldots M\}$. The random variables are connected by edges in $E$ via functions $f_i$ and jointly independent noise variables $U_i$ through $X_i = f_i(X_{pa(i)}, U_i)$ where $X_{pa(i)}$ are $X_i$'s parents in $G$, and directed edges in the graph represent direct causation. The conditionals $P(X_i | X_{pa(i)})$ define the conditional distribution of $X_i$ given its parents.

**Interventions.** Interventions on $X_i$ change the conditional distribution of $P(X_i | X_{pa(i)})$ to a different distribution, hence affecting the outcome of $X_i$. Interventions can be perfect (hard) or imperfect (soft). Hard interventions entirely remove the dependencies of a variable $X_i$ on its parents $X_{pa(i)}$, hence defining the conditional probability distribution of $X_i$ by some $\tilde{P}(X_i)$ rather than $P(X_i | X_{pa(i)})$. A more general form of intervention is the soft intervention, where the intervention changes the effect of the parents of $X_i$ on itself by modifying the conditional distribution from $P_i(X_i | X_{pa(i)})$ to an alternative $\tilde{P}_i(X_i | X_{pa(i)})$.

**Dependency Structure Discovery from Interventions (DSDI).** We evaluate the proposed method under multiple continuous-optimization causal learning frameworks from fused (observational and interventional) data (Bareinboim & Pearl, 2016), one of them being DSDI (Ke et al., 2019). The work of DSDI reformulates the problem of causal discovery from *discrete* data as a continuous optimization problem using neural networks. The framework proposes to learn the causal graph adjacency matrix as a matrix parameter $\gamma$ of a neural network, and is trained using a 3-stage iterative procedure. The first stage involves sampling graphs under the model's current belief in the graph structure and then training functional parameters $\mathcal{P}$ specifying the conditionals of the sampled graphs using observational data. Note how sampling a graph from $\gamma$ can specify element-wise multiplications by 0 or 1 in the first layer of neural networks for the conditionals, to remove unallowed edges, with other network parameters $\mathcal{P}$ applicable to any sampled graph. The next stage is to evaluate the sampled graphs under interventional data and score these graphs accordingly. The final step is to update the learned adjacency matrix with the scores from stage 2. This method performs competitively compared to many other methods. However, all intervention targets in stage 2 of DSDI are random and independent, a strategy that scales poorly to larger graphs. A better approach would have been *active intervention targeting*, elaborated in the next section.

**Differentiable Causal Discovery from Interventional Data (DCDI).** We also consider the work of DCDI (Brouillard et al., 2020), which addresses causal discovery from *continuous* data as a continuous-constrained optimization problem using neural networks to model parameters of Gaus-

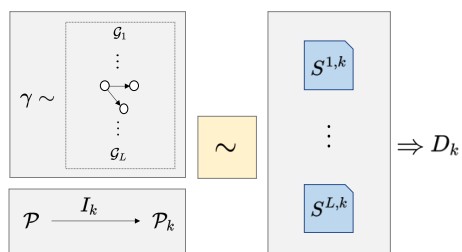
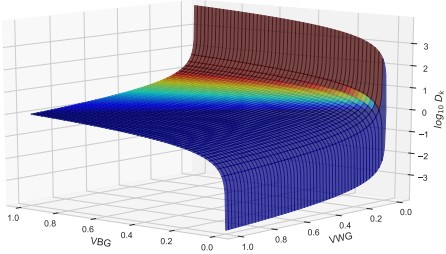

(a) AIT procedure for intervention target $I_k$        (b) Discrepancy score $D_k$ landscape

Figure 1: (a) Procedure: Start by sampling a set of hypothetical graphs $\mathcal{G}$ given the current graph beliefs $\gamma$, and apply an intervention on targets $I_k$ on the functional parameters $\mathcal{P}$ which results in partially altered parameters $\mathcal{P}_k$. Continue by sampling sets of hypothetical interventional experiments under $\mathcal{P}_k$ for every graph in $\mathcal{G}$ and compute the corresponding discrepancy score $D_k$ as a ratio of variance-between-graphs $\mathrm{VBG}_k$ and variance-within-graphs $\mathrm{VWG}_k$. (b) Landscape of the discrepancy score in terms of $\mathrm{VBG}_k$ and $\mathrm{VWG}_k$, in logarithmic scale.

sian distributions or normalizing flows (Rezende & Mohamed, 2015) which represent conditional distributions. Unlike DSDI's iterative training of the structural and functional parameters, DCDI optimizes the causal adjacency matrix and functional parameters jointly over the fused data space. But like DSDI, DCDI uses random and independent interventions.

## 3   ACTIVE INTERVENTION TARGETING

We present a score-based intervention design strategy, called Active Intervention Targeting (AIT), which is applicable to many discrete and continuous optimization formulations of causal structure learning algorithms. Furthermore, we show how our proposed method can be integrated into recent differentiable causal discovery frameworks for guided exploration using interventional data.

**Assumptions.** The proposed method assumes access to a belief state $\gamma$ in the graph structure (e.g., in the form of a distribution over graphs, probabilistic adjacency matrix or a set of hypothetical graphs) and functional parameters $\mathcal{P}$ characterizing the conditional relationships between variables (constrained by the graphs sampled from the belief specified by $\gamma$). The proposed model does not have to assume causal sufficiency per se. However, it inherits the assumptions of the selected base framework, and this may include causal sufficiency depending on the base algorithm of choice. In case the underlying framework can handle unobserved variables and offers a generative method for interventional samples, then our method is also applicable

### 3.1   A SCORE FOR INTERVENTION TARGETING

Given a graph belief state $\gamma$ with its corresponding functional parameters $\mathcal{P}$, and a possible set of intervention targets $I$ (single-node and multi-node intervention targets), we wish to select the most *informative* intervention target(s) $I_{k^*} \in I$ with respect to identifiability of the underlying structure. Such target(s) presumably yield relatively high discrepancies between samples drawn under different hypothesis graphs, indicating larger uncertainty about the target's relation to its parents and/or children.

We thus construct an F-test-inspired score to determine the $I_{k^*}$ exhibiting the high-

---

**Algorithm 1** Active Intervention Targeting (AIT)

**Input:** Functional Parameters $\mathcal{P}$, Graph Belief State $\gamma$,
          Interventional Target Space $I$
**Output:** Intervention Target $I_{k^*}$

1: $\mathcal{G} \leftarrow$ Sample a set of hypothesis graphs from $\gamma$
2: **for** each intervention target $I_k$ in $I$ **do**
3:      $\mathcal{P}_k \leftarrow$ Perform intervention $I_k$ on $\mathcal{P}$
4:      **for each** graph $\mathcal{G}_i$ **in** $\mathcal{G}$ **do**
5:          $S^{k,i} \leftarrow$ Draw samples from $\mathcal{P}_k$ on $\mathcal{G}_i$
6:          $S^{k,i} \leftarrow$ Set variables in $I_k$ to 0
7:      **end for**
8:      Compute: $D_k \leftarrow \dfrac{\sum_i (\mu_i^k - \mu^k)^2}{\sum_i \sum_j (S_j^{k,i} - \mu_i^k)^2}$
9: **end for**
10: Target Intervention $I_{k^*} \leftarrow \arg\max_k (D_k)$

---

est discrepancies between post-interventional sample distributions generated by likely graph structures under fixed functional parameters $\mathcal{P}$. In order to compare sample distributions over different graphs, we distinguish between two sources of variation: Variance *between graphs* ($\mathrm{VBG}$) and variance *within graphs* ($\mathrm{VWG}$). While $\mathrm{VBG}$ characterizes the variance of sample means over multiple

graphs, VWG accounts for the sample variance when a specific graph is fixed. As in DSDI and DCDI, we mask the contribution of the intervened variables $I_k$ to VBG and VWG, and construct our discrepancy score $D$ as a ratio $D = \frac{\text{VBG}}{\text{VWG}}$.

This discrepancy score attains high values for intervention targets of particular interest (see Figure 1b for a landscape visualization). While VBG itself indicates for which intervention targets the model is unsettled about, an extension to the proposed variance ratio enables more control over the region of interest. Given a fixed set of graphs $\mathcal{G}$ and a fixed interventional sample size across all graphs, let us assume a scenario where multiple intervention targets attain high VBG. Assessing VWG allows us to distinguish between two extreme cases: (a) targets with sample populations that exhibit large VWG (b) targets with sample populations that exhibit low VWG. While high VBG in (a) might be induced by an insufficient sample size due to high variance in the interventional distribution itself, (b) clearly indicates high discrepancy between graphs and should be preferentially studied.

**Computational Details.** We begin by sampling a set of graphs $\mathcal{G} = \{\mathcal{G}_i\}$, $i = 1, 2, 3, \ldots$ from our graph structure belief state $\gamma$, however parametrized. This $\mathcal{G}$ will remain fixed for all considered interventions for the current experimental round. Then, we fix an intervention target $I_k$ and apply the corresponding intervention to $\mathcal{P}$, resulting in partially altered functional parameters $\mathcal{P}_k$ where some conditionals have been changed. Next, we draw interventional samples $S^{k,i}$ from $\mathcal{P}_k$ on every graph $\mathcal{G}_i \in \mathcal{G}$ and set variables in $I_k$ to zero to mask of their contribution to the variance. Having collected all samples over the considered graphs for the specific intervention target $I_k$, we compute $\text{VBG}_k$ and $\text{VWG}_k$ as:

$$\text{VBG}_k = \sum_i (\mu_i^k - \mu^k)^2 \qquad \text{and} \qquad \text{VWG}_k = \sum_i \sum_j (S_j^{k,i} - \mu_i^k)^2$$

where $\mu^k$ is a vector of the same dimension as any sample in $S$ and denotes the overall sample-mean of the interventional setting, $\mu_i^k$ the corresponding mean for a specific graph $\mathcal{G}_i$ and $S_j^{k,i}$ is the $j$-th sample of the $i$-th graph configuration. Finally, we construct the discrepancy score $D_k$ of $I_k$ as:

$$D_k \leftarrow \frac{\text{VBG}_k}{\text{VWG}_k}.$$

In contrast to the original definition of the F-Score, we can ignore the normalization constants due to equal group size and degree-of-freedoms. Although the dependence between the variables is apparent from the connected causal structure, we approximate the variance of the multidimensional samples as the trace over the covariance matrix by assuming that the variables are independent. An outline of the method is provided in Algorithm 1.

## 3.2 Two-Phase DAG sampling

Embedding AIT into recent differentiable causal discovery frameworks requires a graph sampler which generates a set of likely graph configurations under the current graph belief state. However, drawing samples from unconstrained graphs (e.g. partially undirected graphs or cyclic directed graphs) is an expensive multi-pass process. Here, we thus constrain our graph sampling space to DAGs. Since most differentiable causal structure learning algorithms learn edge beliefs in the form of a soft-adjacency matrix, we present a scalable, two-stage DAG sampling procedure which exploits structural information of the soft-adjacency matrix beyond independent edge confidences (see Figure 2 for a visual illustration). More precisely, we start by sampling topological node orderings from an iterative refined score and construct DAGs in the constrained space by independent Bernoulli draws over possible edges. We can thus guarantee DAGness by construction and do not have to rely on expensive, non-scalable techniques such as rejection sampling or Gibbs sampling. The overall method is inspired by topological sorting algorithms of DAGs where we iteratively identify nodes with no incoming edges, remove them from the graph and repeat until all nodes are processed.

**Soft-Adjacency.** Given a learnable graph structure $\gamma \in \mathbb{R}^{N \times N}$ of a graph over $N$ variables, the soft-adjacency matrix is given as $\sigma(\gamma) \in [0, 1]^{N \times N}$ such that $\sigma(\gamma_{ij}) \in [0, 1]$ encodes the probabilistic belief in random variable $X_j$ being a direct cause of $X_i$, where $\sigma(x) = (1 + \exp(-x))^{-1}$ denotes the sigmoid function. For the ease of notation, we define $A = \sigma(\gamma)$ and $A^k$ denotes the considered soft-adjacency $\sigma(\gamma)$ at iteration $k$. Note that the shape of $A^k$ changes through the iterations.

**Sample node orderings.** For the iterative root sampling procedure, we start at iteration $k = 0$ with an initial soft-adjacency $A^k = A$ and apply the following routine for $N$ iterations. We take

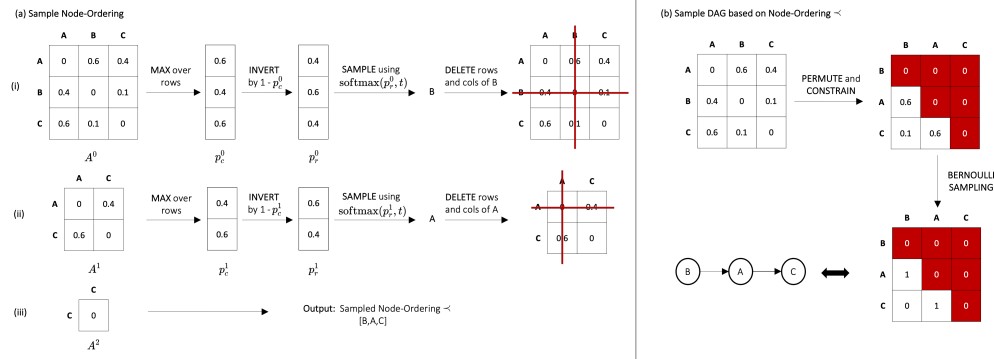

Figure 2: Two-Stage DAG Sampling: Based on a soft-adjacency $\sigma(\gamma)$, we sample a topological node ordering from an iterative refined score which is repeatedly computed until we have processed all nodes of the graph. We proceed by permuting $\sigma(\gamma)$ according to the drawn node ordering and constrain the upper triangular part to ensure DAGness. Finally, we take independent Bernoulli draws of the unconstrained edge beliefs and arrive at a sampled DAG.

the maximum over rows of $A^k$, resulting in a vector of independent probabilities $p_c^k$, where $p_c^k(i)$ denotes the maximal probability of variable $X_i$ being a child of any other variable at the current belief state. After taking the complement $p_r^k = 1 - p_c^k$, we arrive at $p_r^k$ where $p_r^k(i)$ denotes the approximated probability of variable $X_i$ being a root node in the current state. In order to arrive at a normalized distribution to sample a root node, we apply a temperature-scaled softmax:

$$p_s^k(i) = \mathrm{softmax}(p_r^k/t)_i = \frac{\exp\left[p_r^k(i)/t\right]}{\sum_j \exp\left[p_r^k(j)/t\right]}$$

where $t$ denotes the temperature. The introduction of temperature-scaling allows to control the distribution over nodes and account for the entropy of the structural belief. We proceed by sampling a (root) node as $r^k \sim Categorical(p_s^k)$ and delete all corresponding rows and columns from $A^k$ and arrive at a shrinked soft-adjacency $A^{k+1} \in [0,1]^{(N-k-1)\times(N-k-1)}$ over the remaining variables. We repeat the procedure until we have processed all nodes and have a resulting topological node ordering $\prec$ of $[r^0, ..., r^{N-1}]$.

**Sample DAGs based on node orderings.** Given a node ordering $\prec$, we permute the soft-adjacency $A$ accordingly and constrain the upper triangular part by setting values to $0$ to ensure DAGness by construction (as shown in Figure 2). Finally, we sample a DAG by independent Bernoulli draws of the edge beliefs, as proposed in Ke et al. (2019).

### 3.3 APPLICABILITY TO DSDI

Before integrating our method into the DSDI framework, we must choose/design a graph sampler based on DSDI's graph belief characterization and define a sampling routine to generate interventional samples under a given state of the structural and functional parameters.

DSDI offers a learnable graph structure over $N$ variables with $\gamma \in \mathbb{R}^{N\times N}$ such that $\sigma(\gamma) \in [0,1]^{N\times N}$ encodes the soft-adjacency matrix. This formulation naturally suggests the application of the introduced *two-phase DAG sampling* to generate hypothetical DAGs under current beliefs. Under these acyclic graph configurations, one may then apply an intervention to DSDI functional parameters $\mathcal{P}$ and sample data using ancestral sampling.

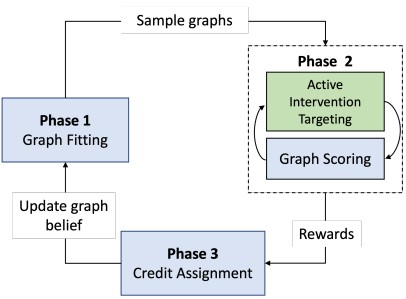

Figure 3: Adapted workflow of DSDI with Active Intervention Targeting

DSDI's architectural choices allow a seamless integration of our proposed active intervention targeting into stage 2 of DSDI, where graphs are evaluated using interventional data. See Figure 3 for an illustrative description of the adapted workflow and §2 for a compact description of the base framework.

### 3.4 Applicability to DCDI

DCDI also offers access to $\gamma \in \mathbb{R}^{N \times N}$ which allows the same setup as with DSDI. In order to generate interventional samples under the hypothetical graphs, we alter the conditionals of the intervened variables and perform ancestral sampling based on the model's learned conditional densities.

Embedding AIT into DCDI allows us to predict an interventional target space instead of relying on random interventional samples chosen out of the full target space. In contrast to the unconstrained target space of the original formulation, we estimate a target space of constrained size using AIT and reevaluate it after a fixed number of gradient steps (see §A.7.1 for technical details).

## 4 Experiments

We evaluate the proposed active intervention targeting mechanism on single-target interventions under two different settings: DSDI (Ke et al., 2019) and DCDI (Brouillard et al., 2020). We investigate the impact of AIT under both settings for identifiability, sample complexity, and convergence behaviour compared to random targeting where the next intervention target is chosen independent of the current evidence. In a further line of experiments, we analyze the targeting dynamics with respect to convergence behaviour and the distribution of target node selections. This section will highlight our results on DSDI while also including key results with respect to DCDI (structural discovery and identifiability). However, further analysis of DCDI results have been shifted to the Appendix.

**Evaluation Setup.** A huge variety of SCMs and their induced DAGs exist, each of which can stress causal structure discovery algorithms in different ways. We perform a systematic evaluation over a selected set of synthetic and non-synthetic SCMs (and datasets). We distinguish between synthetic *structured* graphs and *random* graphs, the latter generated from the Erdős–Rényi (ER) model with varying edge densities (see §A.3 for a detailed description of the setup). For conciseness, we only report results on 15-node graphs in this section for the noise-free synthetic setting for AIT on DSDI and on 10-node graphs for the noisy setting for AIT on DSDI (discrete data). In addition, we report key results on 10-node graphs for AIT on DCDI (continuous data) in the main text and provide further results and ablation studies in Appendix. We complete the setup with the Sachs flow cytometry dataset (Sachs et al., 2005) and the Asia network (Lauritzen & Spiegelhalter, 1988) to evaluate the proposed method on well-known real-world datasets for causal structure discovery. [1]

**Key Findings.** (a) We report strong results for active-targeted structure discovery on both discrete and continuous-valued datasets, outperforming random targeting in all experiments. (b) The proposed intervention targeting mechanism significantly reduces sample complexity with strong benefits for graphs of increasing size and density. (c) The distribution of target selections during graph exploration is strongly connected to the topology of the underlying graph. (d) Our method is capable of identifying informative targets. (e) Undesirable interventions are drastically reduced. (f) When monitoring structured Hamming distance (SHD) throughout the procedure, an "elbow" point appears approximately when the Markov equivalence class (MEC) has been isolated. (g) AIT introduces desirable properties such as improved recovery of erroneously converging edges. (h) AIT significantly improves robustness in noise-perturbed environments.

**Structure discovery: Synthetic datasets.** We evaluate accuracy in terms of Structural Hamming Distance (SHD) (Acid & de Campos, 2003) on a diverse set of synthetic non-linear datasets under both DSDI and DCDI, adopting their respective evaluation setups.

The results of DSDI with AIT are reported in Table 1. DSDI with active intervention targeting outperforms all baselines and DSDI with random intervention targeting over all presented datasets. It enables almost perfect identifiability on all structured graphs of size 15 except for the `full15` graph, and significantly improves structure discovery of random graphs with varying densities. As the size or density of the underlying causal graphs increases, the benefit of the selection policy becomes more apparent (see Figure 4). We also examine the effectiveness of our proposed method for DCDI (Brouillard et al., 2020) on non-linear data from random graphs of size 10. Active Intervention Targeting improves the identification in terms of sample complexity and structural identifiability compared with random exploration (see Figure 5 and §A.7 for further analyses). We observe the

---

[1] The real-world datasets are available through a Creative Commons Attribution-Share Alike License in the bnlearn R package and most baseline implementations are available for Python in the causal discovery toolbox (Kalainathan & Goudet, 2019) with an MIT license. A-ICP is provided at `https://github.com/juangamella/aicp` but without a license.

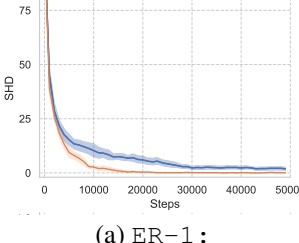 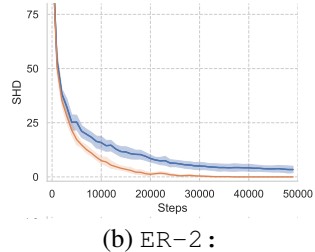 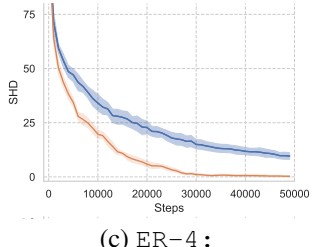

(a) ER-1:  (b) ER-2:  (c) ER-4:

Figure 4: DSDI with active intervention targeting (orange) leads to superior performance over random intervention targeting (blue) on random graphs of size 15. The performance gap becomes more significant with increasing edges density. The plot shows average performance in terms of SHD. Error bands were estimated using 10 random ER graphs per setting.

Table 1: SHD (lower is better) on various 15-variable synthetic datasets. Structured graphs are sorted in ascending order according to their edge density. $^{(*)}$ denotes average SHD over 10 graphs.

| | Structured Graphs | | | | | | Random Graphs | | |
|---|---|---|---|---|---|---|---|---|---|
| | Chain | Collider | Tree | Bidiag | Jungle | Full | ER-1$^{(*)}$ | ER-2$^{(*)}$ | ER-4$^{(*)}$ |
| GES (Chickering, 2002) | 13 | 1 | 12 | 14 | 14 | 69 | 8.3 ($\pm$1.9) | 17.6 ($\pm$4.6) | 39.4 ($\pm$6.7) |
| GIES (Hauser & Bühlmann, 2012) | 13 | 6 | 10 | 17 | 23 | 60 | 10.9 ($\pm$4.2) | 18.1 ($\pm$4.3) | 39.3 ($\pm$5.6) |
| ICP (Peters et al., 2016) | 14 | 14 | 14 | 27 | 26 | 105 | 16.2 ($\pm$3.6) | 31.1 ($\pm$3.4) | 60.1 ($\pm$3.9) |
| A-ICP (Gamella & Heinze-Deml, 2020) | 14 | 14 | 14 | 27 | 26 | 105 | 16.2 ($\pm$3.6) | 31.1 ($\pm$3.4) | 60.1 ($\pm$3.9) |
| NOTEARS (Zheng et al., 2018) | 22 | 21 | 26 | 33 | 35 | 93 | 23.7 ($\pm$4.0) | 35.8 ($\pm$5.2) | 59.5 ($\pm$3.7) |
| DAG-GNN (Yu et al., 2019) | 11 | 14 | 15 | 27 | 25 | 97 | 16.0 ($\pm$3.7) | 30.6 ($\pm$3.4) | 59.7 ($\pm$4.1) |
| DSDI (Random) (Ke et al., 2019) | **0** | **0** | 2 | 3 | 7 | 24 | 1.4 ($\pm$1.6) | 2.1 ($\pm$2.3) | 7.2 ($\pm$2.7) |
| DSDI (AIT) | **0** | **0** | **0** | **0** | **0** | **7** | **0.0 ($\pm$0.0)** | **0.0 ($\pm$0.0)** | **0.0 ($\pm$0.0)** |

clear impact of the targeting mechanisms, which control the order and frequency of interventional targets presented to the model. Further experimental results for DCDI can be found in Appendix.

**Structure discovery: flow cytometry and asia dataset.** While the synthetic datasets systematically explore the strengths and weaknesses of causal structure discovery methods, we further evaluate their capabilities on the real-world flow cytometry dataset (also known as Sachs network)(Sachs et al., 2005) and the Asia network (Lauritzen & Spiegelhalter, 1988) from the BnLearn Repository. DSDI with active intervention targeting outperforms all measured baselines and achieves the same result as random targeting in terms of SHD, but with reduced sample complexity. Despite AIT deviating only

Table 2: SHD (lower is better) on two real-world datasets

| | Sachs | Asia |
|---|---|---|
| GES (Chickering, 2002) | 19 | 4 |
| GIES (Hauser & Bühlmann, 2012) | 16 | 11 |
| ICP (Peters et al., 2016) | 17 | 8 |
| A-ICP (Gamella & Heinze-Deml, 2020) | 17 | 8 |
| NOTEARS (Zheng et al., 2018) | 22 | 14 |
| DAG-GNN (Yu et al., 2019) | 19 | 10 |
| DSDI (Random) (Ke et al., 2019) | **6** | **0** |
| DSDI (AIT) | **6** | **0** |

by 6 undirected edges from the (concensus) ground truth structure of Sachs et al. (Sachs et al., 2005), there is some concern about the correctness of this graph and the different assumptions associated with the dataset (Mooij et al., 2020; Zemplenyi & Miller, 2021). Therefore, perfect identification may not be achievable by any method in practice in the Sachs setting.

**Effect of intervention targeting on sample complexity.** Aside from the significantly improved identification of underlying causal structures, our method allows for a substantial reduction in interventional sample complexity. After reaching the "elbow" point in terms of structural Hamming distance, random intervention targeting requires a fairly long time to converge to a solution within the MEC. In contrast, our proposed technique continues to select informative intervention targets beyond the elbow point and more quickly converges to the correct graph within the MEC. The continued effectiveness of our method directly translates to increased sample-efficiency and convergence speed, and is apparent for all examined datasets (see Figure 4).

**Distribution of intervention targets.** The careful study of the behaviour of the proposed method under our chosen synthetic graphs enable us to reason about the method's underlying dynamics. Analysing the dynamics of intervention targeting reveals that the distribution of target node selections is linked to the topology of the underlying graph. More specifically, the number of selections of a given target node strongly correlates with its out-degree and number of descendants in the underlying ground-truth graph structure (see Figure 7). That our method prefers interventions on nodes with

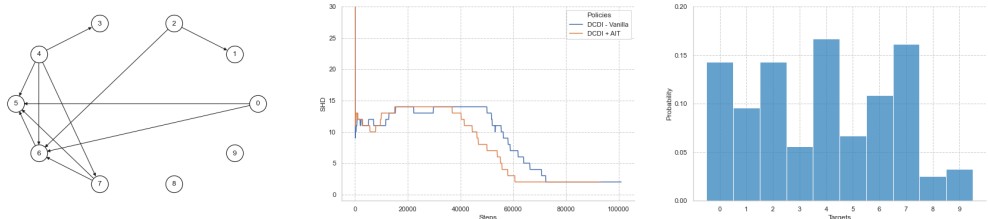

Figure 5: AIT-guided DCDI (orange) allows a more rapid discovery of the causal structure compared to DSDI relying on random interventions (blue). The distribution of selected intervention targets shows its correlated connection to the topology of the underlying graph where nodes of greater impact on the overall system are preferentially studied and nodes without children are rarely chosen.

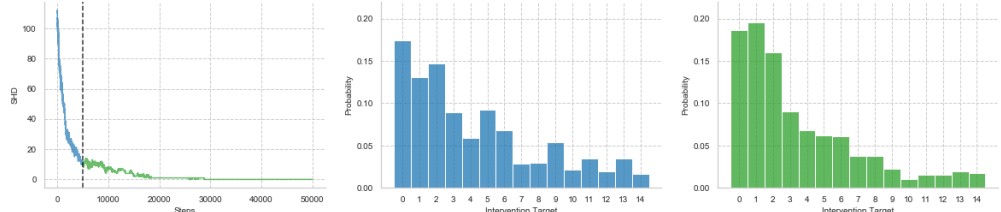

Figure 6: DSDI: Dynamics and target distribution of AIT for a structured `jungle` graph of size 15. The graphs' nodes are sorted in topological order, root node first. The graph is binary-tree-like with 4 levels. For the dense `jungle15`, the multi-level structure characteristic of the tree-like graph is readily apparent even before the elbow point. Nodes without children are very rarely chosen.

greater (downstream) impact on the overall system can be most clearly observed in the distribution of target selection on the example of the synthetic `jungle` graph in Figure 6.

**Selection of informative targets.** Apart our strong results in the discovery of the underlying causal graph, we demonstrate AIT's general ability of detecting informative intervention targets in a careful designed empirical study in Appendix §A.6.7.

**Reduction of undesirable interventions.** An intervention destroys the original causal influence of other variables on the intervened target variable $I_k$, so its samples cannot be used to determine the causal parents of $I_k$ in the undisturbed system. Therefore, if a variable without children is detected, interventions upon it should be avoided since they effectively result in redundant observational samples of the remaining variables that are of no benefit for causal structure discovery. Active intervention targeting leads to the desirable property that interventions on such variables are drastically reduced (see Figure 5 and 6).

**Identification of Markov equivalence class.** Investigating the evolution of the intervention target distribution over time reveals that the causal discovery seems to be divided into two phases of exploration: Phase 1 lasts until the elbow point in terms of SHD, and Phase 2 from the elbow point until convergence (see Figure 4). We observed over multiple experiments that phase 1 tends to quickly discover the underlying skeleton (removing superfluous connections while keeping some edges undirected), until a belief state $\gamma_{elbow}$ is reached representing a MEC, or a class of graphs very close to a MEC. Phase 2 is predominantly operating on the partially directed skeleton and directed on the remaining edges.

**Recovery of erroneously converging edges.** Recovery of incorrectly-converging edges critically depends on adapting the order of interventions, which a random intervention policy does not. In sharp contrast, intervention targeting significantly promotes early recovery from incorrect assignment of an edge. In contrast, the observed edge dynamics and the corresponding graph belief states indicate that the random policy can lock itself into unfavorable belief states from which recovery is extremely difficult, while AIT provides an escape hatch throughout learning.

**AIT improves robustness in noise-perturbed environments.** Considering that noise significantly impairs the performance of causal discovery, we examine the performance of active intervention targeting in noise-perturbed environments with respect to SHD and convergence speed and compare it with random intervention targeting. We conduct experiments under different noise levels in the setting of binary data generated from structured and random graphs of varying density. A noise level

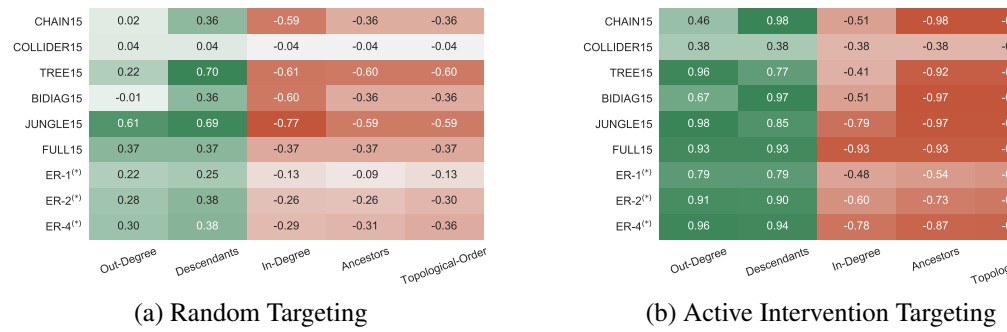

(a) Random Targeting                      (b) Active Intervention Targeting

Figure 7: Correlation scores between the number of individual target selections and different topological properties of those targets. AIT shows strong correlations with the measured properties over all graphs, which indicates a controlled discovery of the underlying structure through preferential targeting of nodes with greater (downstream) impact on the overall system.

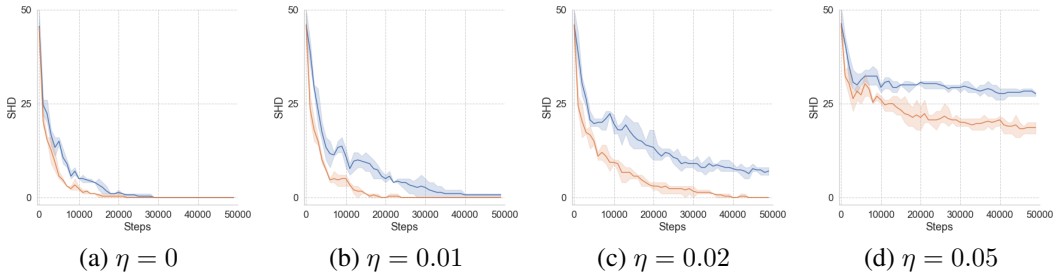

(a) $\eta = 0$           (b) $\eta = 0.01$           (c) $\eta = 0.02$           (d) $\eta = 0.05$

Figure 8: Convergence Behaviour in terms of SHD for random ER-4 graphs over 10 variables under different noise levels $\eta$, where Active Intervention Targeting (orange) clearly outperforms Random Targeting (blue) over all noise levels. The performance gap becomes of larger magnitude as the noise level increases. Error bands were estimated using 3 random ER graphs per setting.

$\eta$ denotes the probability of flipping a random variable and applying it to all measured variables of observational and interventional samples. Through all examined settings, we observe that active intervention targeting significantly improves identifiability in contrast to random targeting (see §A.6.6 for detailed results). Active intervention targeting perfectly identifies all structured graphs, except for the collider and full graph, up to a noise level of $\eta = 0.05$, i.e. where every 20th variable is flipped. The observed performance boost is even more noticeable in the convergence speed, as shown in Fig. 8 for ER-4 graphs spanning over 10 variables. While the convergence-gap gets more significant with an increasing noise level, random targeting does not converge to the ground-truth graphs for a noise level higher than $\eta = 0.02$. In contrast, AIT still converges to the correct graph and shows even a convergence tendency for $\eta = 0.05$. These findings support our observation from different experiments that active intervention targeting leads to a more controlled and robust graph discovery. Further experimental results in noise-perturbed environments can be found in Appendix.

## 5 CONCLUSION

Promising results have driven the recent surge of interest in continuous optimization methods for Bayesian network structure learning from observational and interventional data. In this work, we propose an active learning method to choose interventions that help to identify the underlying graph efficiently in the setting of differentiable causal discovery. We show via a detailed empirical study that active intervention targeting not only improves sample efficiency but also the identification of the underlying causal structures compared to random targeting of interventions.

While our method shows significant improvements with respect to sample efficiency and graph recovery over existing methods across multiple noise-free and noise-perturbed datasets, the number of interventions is not yet optimal (Atkinson & Fedorov, 1975; Eberhardt et al., 2012) and can potentially be reduced in future work. Further, in this work, the interventional samples were presented to the evaluated frameworks according to a fixed learning schema (e.g. fixed number of samples for evaluated interventions in graph scoring). It would be interesting to see if the information discovered by AIT could be used for a more adaptive learning procedure to further improve sample efficiency.

## 6 REPRODUCIBILITY

We encourage the reproduction and extension of our presented work by disclosing all relevant information. From a methodological point of view, we provide detailed information including algorithm outlines for the proposed score-based intervention targeting mechanism in section section 3.1 and similarly for the proposed two-stage DAG sampling technique in section 3.2 and appendix A.2. Regarding the embedding of the proposed AIT framework into existing differentiable causal discovery frameworks, we introduce the seamless embedding of our proposed method into DSDI (Ke et al., 2019) in section 3.3 and discuss the embedding including relevant structural changes of DCDI (Brouillard et al., 2020) in section 3.3 and appendix A.7.1. Moreover, we report all hyperparameters in appendix A.5 and provide a link to an anonymous repository for reviewers and area chairs once the discussion forums open. The fully documented code will be soon publicly released to the entire audience.

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

# A APPENDIX

## CONTENTS

## A.1 RELATED WORK

Causal induction can use either observational and (or) interventional data. With purely observational data, the causal graph is only *identifiable* up to a Markov equivalence class (MEC) (Spirtes et al., 2000), *interventions* are needed in order to identify the underlying causal graph (Eberhardt & Scheines, 2007). Our work focuses on causal induction from interventional data.

**Causal Structure Learning.** There exists several approaches for causal induction from interventional data: score-based, constraint-based, conditional independence test based and continuous optimization. We refer to (Heinze-Deml et al., 2018; Vowels et al., 2021) for recent overviews. While most algorithms perform heuristic, guided searches through the discrete space of DAGs, Zheng et al. (2018) reformulates it as a continuous optimization problem constrained to the zero level set of the adjacency matrix exponential. This important result has driven recent work in the field and showed promising results (Kalainathan et al., 2018; Yu et al., 2019; Ng et al., 2019; Lachapelle et al., 2020; Zheng et al., 2020; Zhu et al., 2020). Due to the limitations of purely observational data, Ke et al. (2019) and Brouillard et al. (2020) extend the continuous optimization framework to make use of interventional data. Lippe et al. (2021) scales in a concurrent work with ours the work of (Ke et al., 2019) to higher dimensions by splitting structural edge parameters in separate orientation and likelihood parameters and leveraging it in an adapted gradient formulation with lower variance. In contrast to (Brouillard et al., 2020; Ke et al., 2019) and our work, they require interventional data on every variable.

**Active Causal Structure Learning.** Interventions are usually hard to perform and in some cases even impossible (Peters et al., 2017). Minimizing the number of interventions performed is desirable. Active causal structure learning addresses this problem, and a number of approaches have been proposed in the literature. These approaches can be divided into those that select intervention targets using graph-theoretic frameworks, and those using Bayesian methods and information gain.

Graph-theoretic frameworks usually proceed from a pre-specified MEC or CPDAG (completed partially directed acyclic graph) and either investigate special graph substructures (He & Geng, 2008) such as cliques (Eberhardt, 2012; Squires et al., 2020), trees (Greenewald et al., 2019), or they prune and orient edges until a satisfactory solution is reached (Ghassami et al., 2018; 2019; Hyttinen et al., 2013), perhaps under a cost budget (Kocaoglu et al., 2017a; Lindgren et al., 2018). Their chief limitation is that an incorrect starting CPDAG can prevent reaching the correct graph structure even with an optimal choice of interventions.

The other popular set of techniques involve sampling graphs from the posterior distribution in a Bayesian framework using MCMC and then selecting the interventions which maximize the information gain on discrete (Murphy, 2001; Tong & Koller, 2001) or Gaussian (Cho et al., 2016) variables. The drawbacks of these techniques are poor scaling and the difficulty of integrating them with non-Bayesian methods, except perhaps by bootstrapping (Agrawal et al., 2019).

In contrast to existing work, our base frameworks do not start from a pre-specified MEC or CPDAG and existing graph-theoretical approaches are hence not applicable unless we pre-initalize them with a known skeleton. However, in the case we offer access to a predefined structure in the form of a MEC or CPDAG, a previously directed edge is likely to be inverted during the ongoing process which contradicts with the underlying assumptions of existing approaches. Further, we build atop non-Bayesian frameworks and are therefore limited in applying methods based on information gain which require access to a posterior distribution over graph structures. While bootstrapping would allow us to approximate the posterior distribution over graph structures in our non-Bayesian setting, it is not guaranteed to achieve full support over all graphs since the support is limited to graphs estimated in the bootstrap procedure (Agrawal et al., 2019). Furthermore, the computational burden of bootstrap would limit us in scaling to graphs of larger size.

## A.2 Two-Stage DAG Sampling

### A.2.1 Algorithm Outline

We present an outline of the proposed two-stage DAG sampling procedure which exploits structural information of the soft-adjacency beyond independent edge confidences. The routine is based on a graph belief state $\gamma$ where $\sigma(\gamma)$ denotes a soft-adjacency characterization. We start by sampling topological node orderings from an iterative refined score and construct DAGs in the constrained space by independent Bernoulli draws over possible edges. We can therefore guarantee DAGness by construction.

The temperature parameter $t > 0$ of the temperature-scaled softmax can be used to account for the entropy of the graph belief state. However, in the general setting we suggest to initialize the parameter to $t = 0.1$. Note that initializing $t \to 0$ results in always picking the maximizing argument and $t \to \infty$ results in an uniform distribution.

---

**Algorithm 2** Two-Stage DAG Sampling

---

**Input:** Graph Belief State $\sigma(\gamma)$ in the form of a soft-adjacency matrix
**Output:** DAG Adjacency Matrix $A_{Dag}$

$\triangleright$ **Phase 1:** Sample Node Ordering $\prec$

1: $A^0 \leftarrow \sigma(\gamma)$
2: $nodes \leftarrow [0, ..., N-1]$
3: **for** $k = 0$ to $N-1$ **do**
4:      $p_c^k(i) \leftarrow \max A^k[i, :]$
5:      $p_r^k(i) \leftarrow 1 - p_c^k(i)$
6:      $p_s^k(i) \leftarrow \frac{\exp[p_r^k(i)/t]}{\sum_j \exp[p_r^k(j)/t]}$
7:      $r_k \leftarrow nodes[idx^k]$ where $idx^k \sim Categorical(p_s^k)$
8:      Remove $r_k$ from $nodes$
9:      $A^{k+1} \leftarrow A^k[nodes, nodes]$
10: **end for**
11: $\prec = [r^0, ..., r^{N-1}]$

$\triangleright$ **Phase 2:** Sample DAG based on node ordering $\prec$

12: $A_{Perm} \leftarrow$ Permute $\sigma(\gamma)$ according to $\prec$
13: $A_{Perm} \leftarrow$ Constrain upper diagonal part by setting values to 0
14: $A_{Ber} \leftarrow$ Bernoulli($A_{Perm}$)
15: $A_{Dag} \leftarrow$ Apply inverse permutation of $\prec$ to $A_{Ber}$

---

### A.2.2 Connection to Plackett-Luce distribution (Luce, 1959; Plackett, 1975)

Our proposed node ordering sampling routine can be regarded as an extension of the Placket-Luce distribution over node permutations. In contrast, we refine scores in an iterative fashion rather than setting them apriori as we account for previously drawn nodes to estimate the probability of a node being the root node in the current iteration.

### A.3 Experimental Setup

A huge variety of SCMs and their induced DAGs exist, each of which can stress causal structure discovery algorithms in different ways. In this work, We perform a systematic evaluation over a selected set of synthetic and non-synthetic SCMs. We distinguish between discrete (based on DSDI (Ke et al., 2019)) or continuous (based on DCDI (Brouillard et al., 2020)) valued random variables. Through all experiment, we limit us to 1000 samples per intervention.

#### A.3.1 Synthetic Datasets

**Graph Structure.** We adopt the structured graphs (see Fig. 9) proposed in the work of DSDI (Ke et al., 2019) as they adequately represent topological diversity of possible DAGs in a compact fashion. They can be split up in a set of graphs without cycles in the undirected skeletons, and one group with cycles. Extending the setup with random graphs with varying edge densities, generated from the Erdős–Rényi (ER) model, allows us to assess the generalized performance of the proposed method from sparse to dense DAGs.

**Discrete Data Generation.** We adopt the generative setup of DSDI (Ke et al., 2019) and model the SCMs using two-layer MLPs with Leaky ReLU activations between layers. For every variable $X_i$, a seperate MLP models the conditional relationship $P(X_i|X_{pa(i)})$. The MLP parameters are initialized orthogonally within the range of $[-2.5, 2.5]$ and biases uniformly in the range of $[-1.1, 1.1]$.

**Continuous Data Generation.** For the evaluation of the adapted DCDI framework, we adopt their generative setup as described in (Brouillard et al., 2020) and use the existing non-linear datasets.

**Graphs with acyclic skeletons:**

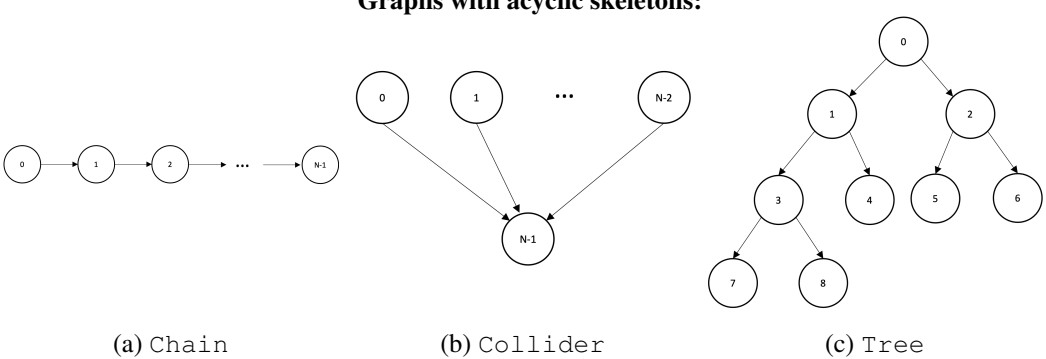

(a) Chain      (b) Collider      (c) Tree

**Graphs with cyclic skeletons:**

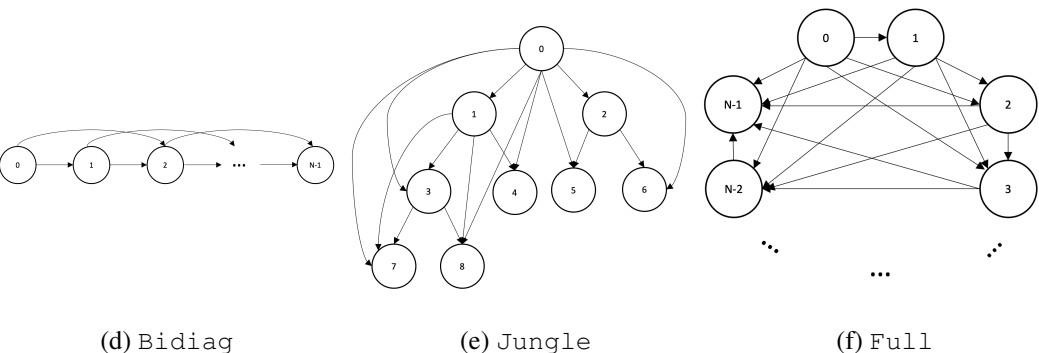

(d) Bidiag      (e) Jungle      (f) Full

Figure 9: Visualization of Structured Graphs as proposed in Ke et al. (2019) - adapted illustration

### A.3.2 REAL-WORLD DATASETS

Besides the many synthetic graphs, we evaluate our method on real-world datasets provided by the BnLearn data repository. Namely on the Asia (Lauritzen & Spiegelhalter, 1988) and the Sachs (Sachs et al., 2005) datasets (see Fig. 10 for a visualization of their underlying ground-truth structure). Sachs (Sachs et al., 2005) represents a systems biology dataset which exhibits non-linearity, confounding and complex structure.

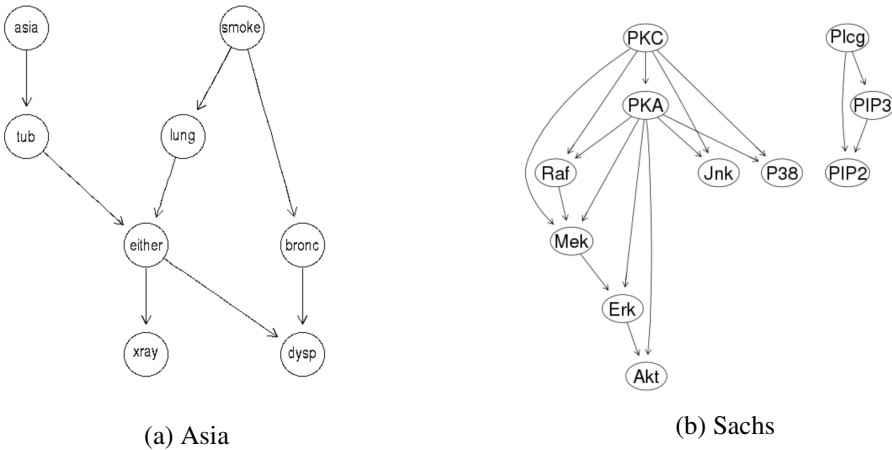

(a) Asia

(b) Sachs

Figure 10: Ground-truth structure of the evaluated real-world datasets provided by the BnLearn data repository - Illustration from: `https://www.bnlearn.com/bnrepository/discrete-small.html`

### A.4 AVAILABILITY OF USED (EXISTING) ASSETS

**Base Frameworks.**

- DSDI (Ke et al., 2019): `https://github.com/nke001/causal_learning_unknown_interventions`
- DCDI (Brouillard et al., 2020): `https://github.com/slachapelle/dcdi`

**Baseline Methods.**

- GES (Chickering, 2002) and GIES (Hauser & Bühlmann, 2012): `www.github.com/FenTechSolutions/CausalDiscoveryToolbox` (Kalainathan & Goudet, 2019)
- ICP (Peters et al., 2016): `https://github.com/juangamella/aicp`
- A-ICP (Gamella & Heinze-Deml, 2020): `https://github.com/juangamella/aicp`
- NOTEARS (Zheng et al., 2018): `https://github.com/xunzheng/notears`
- DAG-GNN (Yu et al., 2019): `https://github.com/fishmoon1234/DAG-GNN`

**Datasets.**

- BnLearn Data Repository: `https://www.bnlearn.com/bnrepository/`

## A.5 Hyper-Parameters

We used a similar set of hyperparameters for our AIT + DSDI and AIT + DCDI models as those used in the original paper (Ke et al., 2019; Brouillard et al., 2020). The specific hyperparamters we used are stated as follows.

**DSDI.**

Table 3: Hyperparameters for DSDI including the corresponding AIT parameters

| | |
|---|---|
| Number of iterations | 1000 |
| Batch size | 256 |
| Sparsity Regularizer | 0.1 |
| DAG Regularizer | 0.5 |
| Functional parameter training iterations | 10000 |
| Number of interventions per phase 2 | 25 |
| Number of data batches for scoring | 10 |
| Number of graph configurations for scoring | |
| - Graph Size 5: | 10 |
| - Graph Size 10: | 20 |
| - Graph Size 15 | 40 |
| AIT: | |
| - Number of graph configurations | 100 |
| - Number of interventional samples per graph & target | 256 |

**DCDI.**

Table 4: Hyperparameters for DCDI including the corresponding AIT parameters

| | |
|---|---|
| $\mu_0$ | $10^{-8}$ |
| $\gamma_0$ | 0 |
| $\eta$ | 2 |
| $\delta$ | 0.9 |
| Augmented Lagrangian Thresh | $10^{-8}$ |
| Learning rate | $10^{-3}$ |
| Nr. of hidden units | 16 |
| Nr. of hidden layers | 2 |
| AIT: | |
| - Number of graph configurations | 100 |
| - Number of interventional samples per graph & target | 256 |

## A.6 Discrete Setting: Additional Experiments / Results

In this section, we show further results and visualizations of experiments on discrete data and single-target interventions in various settings (such as graphs of varying size, noise-free vs. noise-perturbed, limited intervention targets). All experiments are based on the framework DSDI.

### A.6.1 Evaluation (SHD) on graphs of varying size and density

Table 5: SHD (lower is better) on various 5-variable synthetic datasets. Structured graphs are sorted in ascending order according to their edge density. $^{(*)}$ denotes average SHD over 10 random graphs., †ER-2 graphs on 5 results in the full5 graph and ER-4 graphs on 5 node graphs are non-existing

| | Structured Graphs | | | | | | Random Graphs | | |
|---|---|---|---|---|---|---|---|---|---|
| | Chain | Collider | Tree | Bidiag | Jungle | Full | ER-1$^{(*)}$ | ER-2$^{(*)}$ | ER-4$^{(*)}$ |
| GES (Chickering, 2002) | 3 | 0 | 4 | 6 | 4 | 9 | 4.3 (±1.0) | † | † |
| GIES (Hauser & Bühlmann, 2012) | 3 | 4 | 2 | 6 | 5 | 10 | 4.7 (±1.6) | † | † |
| ICP (Peters et al., 2016) | 4 | 4 | 4 | 7 | 6 | 10 | 5.4 (±1.4) | † | † |
| A-ICP (Gamella & Heinze-Deml, 2020) | 4 | 4 | 4 | 7 | 6 | 10 | 5.4 (±1.4) | † | † |
| NOTEARS (Zheng et al., 2018) | 5 | 3 | 6 | 5 | 7 | 9 | 6.1 (±1.7) | † | † |
| DAG-GNN (Yu et al., 2019) | 4 | 4 | 3 | 4 | 6 | 9 | 5.1 (±1.4) | † | † |
| DSDI (Random) (Ke et al., 2019) | 0 | 0 | 0 | 0 | 0 | 0 | 0.0 (±0.0) | † | † |
| DSDI (AIT) | 0 | 0 | 0 | 0 | 0 | 0 | 0.0 (±0.0) | † | † |

Table 6: SHD (lower is better) on various 10-variable synthetic datasets. Structured graphs are sorted in ascending order according to their edge density. $^{(*)}$ denotes average SHD over 10 random graphs.

| | Structured Graphs | | | | | | Random Graphs | | |
|---|---|---|---|---|---|---|---|---|---|
| | Chain | Collider | Tree | Bidiag | Jungle | Full | ER-1$^{(*)}$ | ER-2$^{(*)}$ | ER-4$^{(*)}$ |
| GES (Chickering, 2002) | 9 | 2 | 6 | 8 | 10 | 35 | 7.0 (±1.6) | 10.7 (±3.8) | 26.7 (±2.9) |
| GIES (Hauser & Bühlmann, 2012) | 12 | 6 | 13 | 16 | 9 | 20 | 12.2 (±5.1) | 14.1 (±4.7) | 26.1 (±4.4) |
| ICP (Peters et al., 2016) | 9 | 9 | 9 | 17 | 16 | 45 | 10.6 (±2.5) | 20.7 (±3.3) | 39.8 (±1.9) |
| A-ICP (Gamella & Heinze-Deml, 2020) | 9 | 9 | 9 | 17 | 16 | 45 | 10.6 (±2.5) | 20.7 (±3.3) | 39.8 (±1.9) |
| NOTEARS (Zheng et al., 2018) | 13 | 16 | 12 | 21 | 21 | 42 | 16.4 (±3.4) | 22.9 (±2.9) | 36.6 (±2.6) |
| DAG-GNN (Yu et al., 2019) | 8 | 7 | 6 | 15 | 13 | 38 | 10.3 (±2.8) | 20.1 (±3.5) | 38.4 (±1.9) |
| DSDI (Random) (Ke et al., 2019) | 0 | 0 | 0 | 0 | 0 | 0 | 0.0 (±0.0) | 0.0 (±0.0) | 0.0 (±0.0) |
| DSDI (AIT) | 0 | 0 | 0 | 0 | 0 | 0 | 0.0 (±0.0) | 0.0 (±0.0) | 0.0 (±0.0) |

Table 7: SHD (lower is better) on various 15-variable synthetic datasets. Structured graphs are sorted in ascending order according to their edge density. $^{(*)}$ denotes average SHD over 10 random graphs.

| | Structured Graphs | | | | | | Random Graphs | | |
|---|---|---|---|---|---|---|---|---|---|
| | Chain | Collider | Tree | Bidiag | Jungle | Full | ER-1$^{(*)}$ | ER-2$^{(*)}$ | ER-4$^{(*)}$ |
| GES (Chickering, 2002) | 13 | 1 | 12 | 14 | 14 | 69 | 8.3 (±1.9) | 17.6 (±4.6) | 39.4 (±6.7) |
| GIES (Hauser & Bühlmann, 2012) | 13 | 6 | 10 | 17 | 23 | 60 | 10.9 (±4.2) | 18.1 (±4.3) | 39.3 (±5.6) |
| ICP (Peters et al., 2016) | 14 | 14 | 14 | 27 | 26 | 105 | 16.2 (±3.6) | 31.1 (±3.4) | 60.1 (±3.9) |
| A-ICP (Gamella & Heinze-Deml, 2020) | 14 | 14 | 14 | 27 | 26 | 105 | 16.2 (±3.6) | 31.1 (±3.4) | 60.1 (±3.9) |
| NOTEARS (Zheng et al., 2018) | 22 | 21 | 26 | 33 | 35 | 93 | 23.7 (±4.0) | 35.8 (±5.2) | 59.5 (±3.7) |
| DAG-GNN (Yu et al., 2019) | 11 | 14 | 15 | 27 | 25 | 97 | 16.0 (±3.7) | 30.6 (±3.4) | 59.7 (±4.1) |
| DSDI (Random) (Ke et al., 2019) | 0 | 0 | 2 | 3 | 7 | 24 | 1.4 (±1.6) | 2.1 (±2.3) | 7.2 (±2.7) |
| DSDI (AIT) | 0 | 0 | 0 | 0 | 0 | 7 | 0.0 (±0.0) | 0.0 (±0.0) | 0.0 (±0.0) |

A.6.2   EVALUATION OF CONVERGENCE SPEED ON GRAPHS OF VARYING SIZE AND DENSITY

While we have shown the effectiveness of AIT on random ER graphs of size 15 in §4, we observe similar effects on ER graphs of size 10 (see Figure 11). Overall, the results indicate a greater impact of our proposed targeting mechanisms on graphs of bigger size compared to random intervention targeting which poorly scales to graphs of larger size.

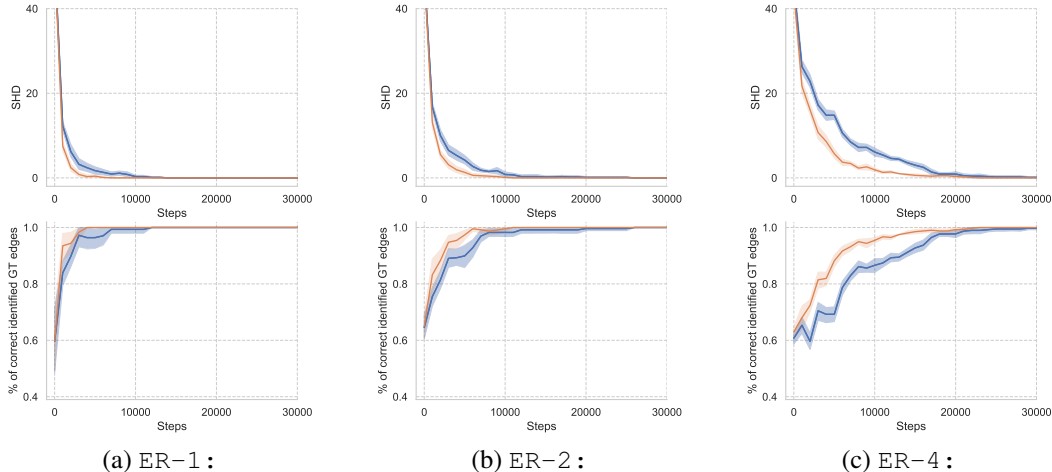

Figure 11: DSDI with AIT (orange) leads to superior performance over random intervention targeting (blue) on **random graphs of size 10** of varying edge densities. Error bands were estimated using 10 random ER graphs per setting.

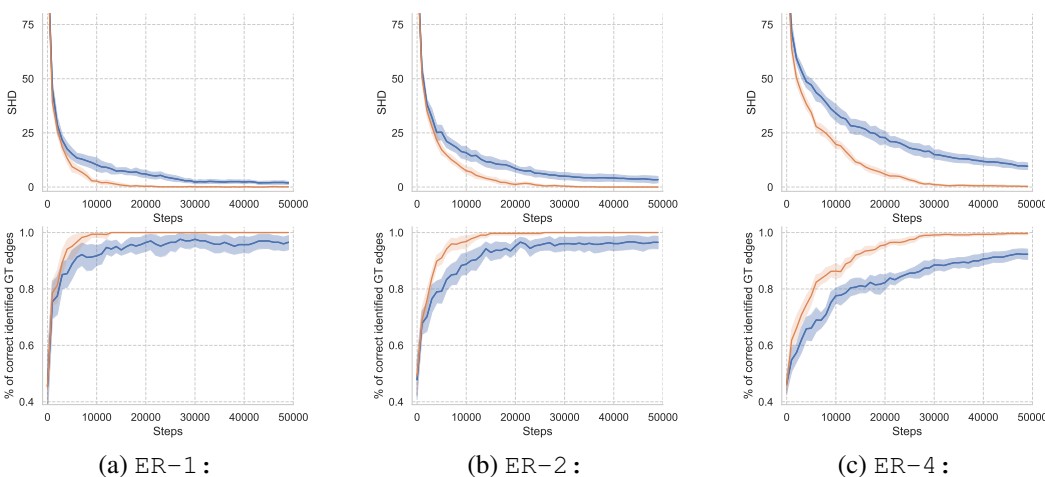

Figure 12: DSDI with AIT (orange) leads to superior performance over random intervention targeting (blue) on **random graphs of size 15** of varying edge densities. Error bands were estimated using 10 random ER graphs per setting.

### A.6.3 TARGET SELECTION ANALYSIS FOR GRAPHS OF VARYING SIZE AND DENSITY

We evaluate the distribution of target node selections over multiple DAGs of varying size to investigate the behaviour of our proposed method. Over all performed experiments, our method prefers interventions on nodes with greater (downstream) impact on the overall system, i.e. nodes of higher topological rank in the underlying DAG.

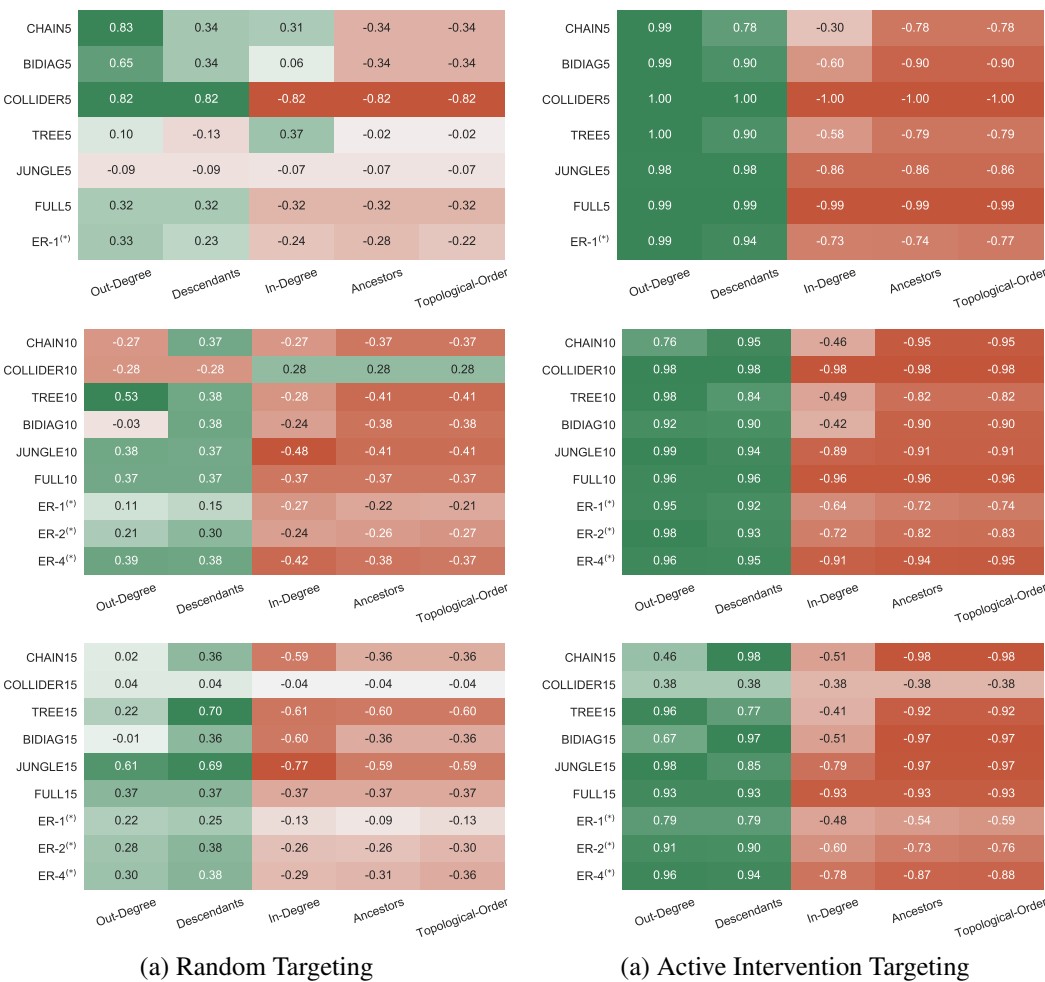

(a) Random Targeting      (a) Active Intervention Targeting

Figure 13: Correlation scores over graphs of varying size and density between the number of individual target selections and different topological properties of those targets. AIT shows strong correlations with the measured properties over all graphs, which indicates a controlled discovery of the underlying structure through preferential targeting of nodes with greater (downstream) impact on the overall system.

### A.6.4 Visualization of target distribution on structured graphs of size 5

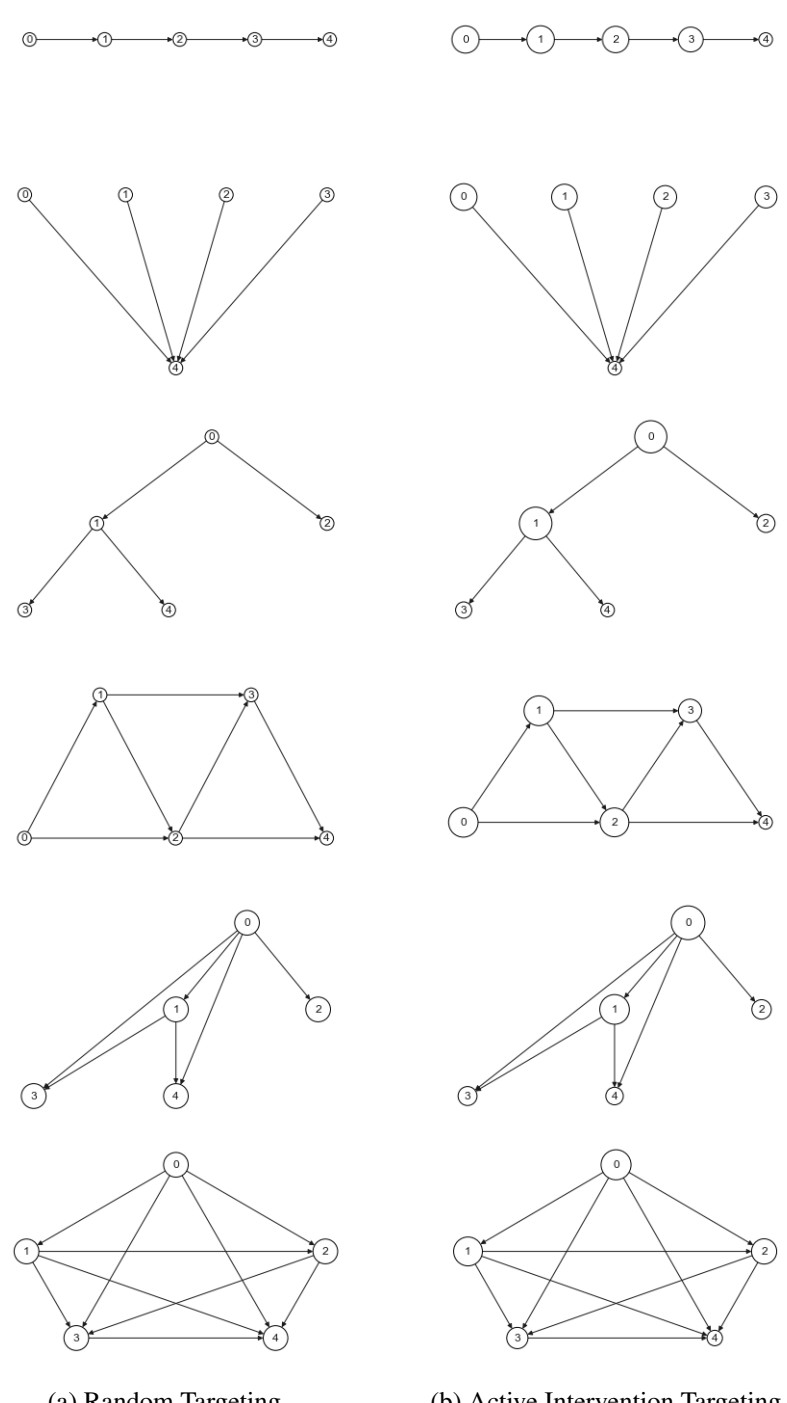

(a) Random Targeting      (b) Active Intervention Targeting

Figure 14: Visualization of target selection on structured graphs of size 5 - bigger node size denotes more selection of the node. While Random Targeting acts as we expect and selects every node an uniform amount, AIT prefers targeting of nodes with greater (downstream) impact on the overall system, i.e. nodes of higher topological order.

### A.6.5    EXTENDED ANALYSIS OF EDGE DYNAMICS

We show all edge dynamics of all structured graphs over 15 variables and compare the dynamics of random targeting to active intervention targeting in a noise-free setting where we have access to all possible single-target interventions.

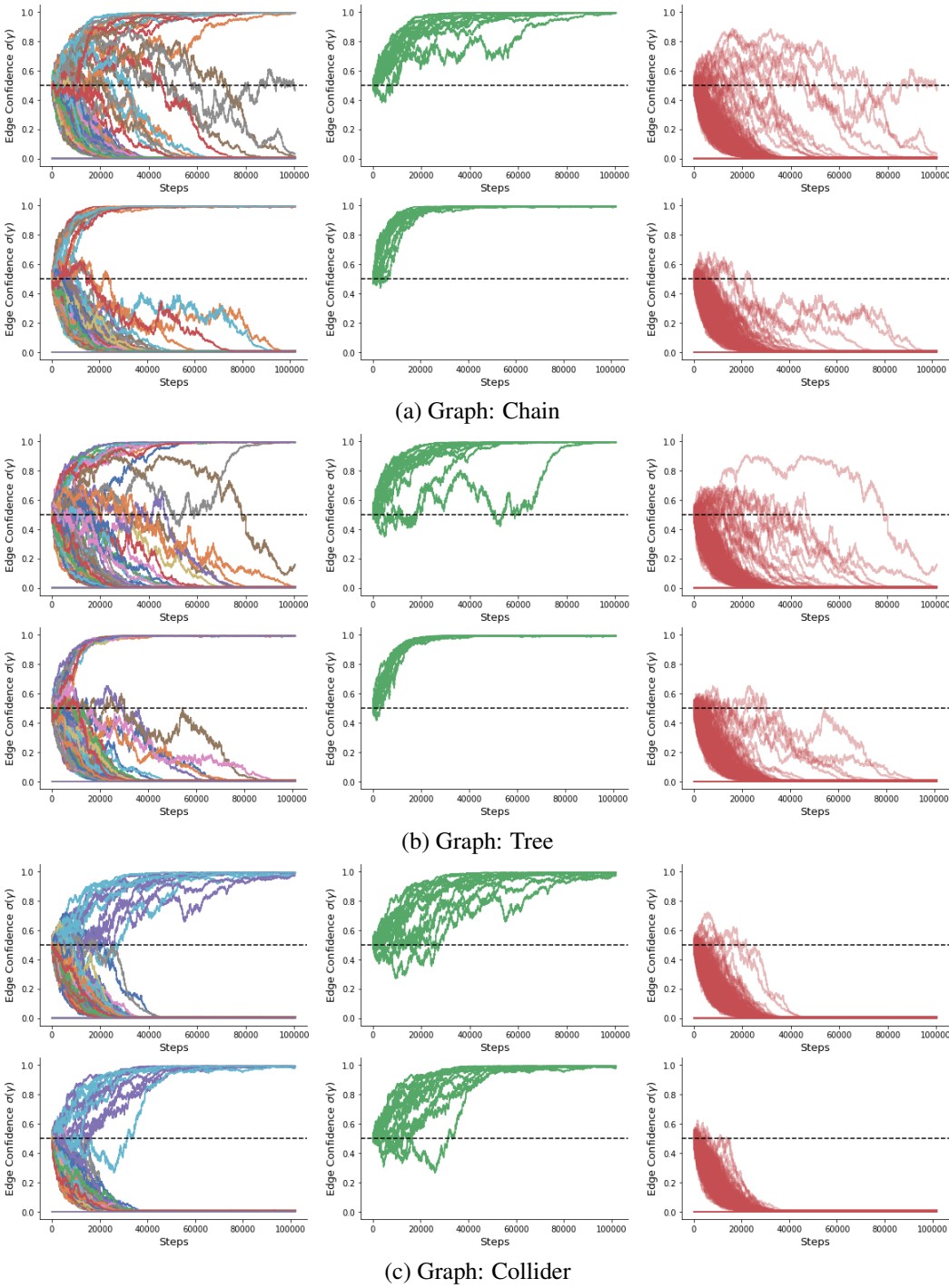

(a) Graph: Chain

(b) Graph: Tree

(c) Graph: Collider

Figure 15: Edge Dynamics of the examined structured graphs spanning over 15 variables - Part 1: The upper part shows the dynamics of random targeting and the lower of active intervention targeting.

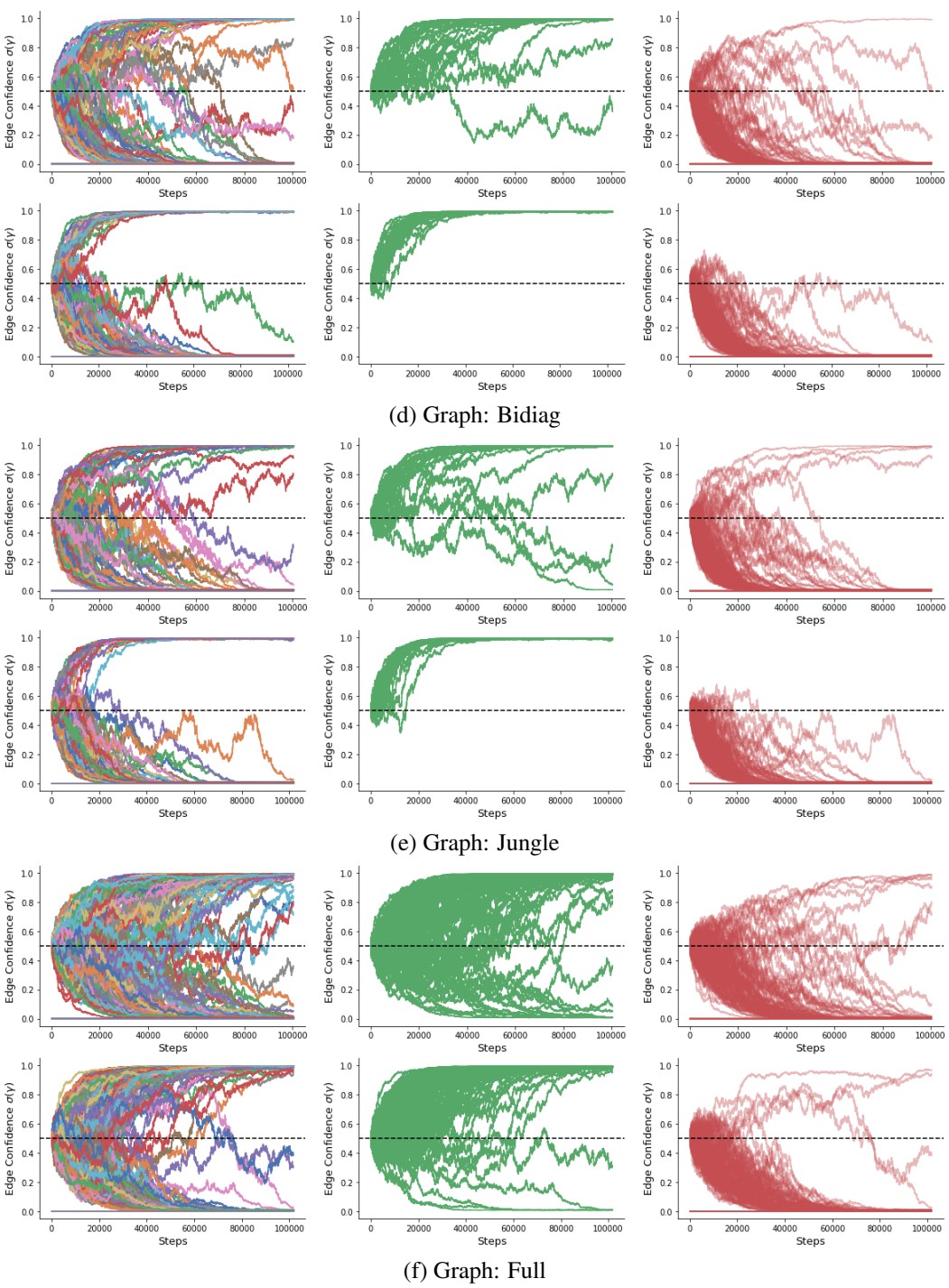

(d) Graph: Bidiag

(e) Graph: Jungle

(f) Graph: Full

Figure 16: Edge Dynamics of the examined structured graphs spanning over 15 variables - Part 2: The upper part shows the dynamics of random targeting and the lower of active intervention targeting.

### A.6.6 IMPROVED ROBUSTNESS WITH DSDI+AIT IN NOISE PERTURBED ENVIRONMENTS

While section §4 highlights our key findings in noise-perturbed systems, we examine the impact of AIT in noise perturbed environments more thoroughly in this section. Therefore, we systematically analyze experiments under different noise levels in the setting of binary data generated from random graphs of varying densities. A noise level $\eta$ denotes the probability of flipping a random variable and apply it to all measured variables of observational and interventional samples.

Evaluating convergence on various ER graphs of varying densities over 10 variables under different noise levels reveals that the impact of AIT becomes of larger magnitude as the density of the graph and the noise level increases.

Table 8: Performance evaluation (SHD) under different noise level $\eta$ for structured and random graphs $^{(*)}$ denotes average SHD over 3 random graphs.

| | | Chain10 | Collider10 | Tree10 | Bidiag10 | Jungle10 | Full10 | ER-1$^{(*)}$ | ER-2$^{(*)}$ | ER-4$^{(*)}$ |
|---|---|---|---|---|---|---|---|---|---|---|
| $\eta = 0.0$ | Random | 0 | 0 | 0 | 0 | 0 | 0 | 0.0 ($\pm$0.0) | 0.0 ($\pm$0.0) | 0.0 ($\pm$0.0) |
| | AIT | 0 | 0 | 0 | 0 | 0 | 0 | 0.0 ($\pm$0.0) | 0.0 ($\pm$0.0) | 0.0 ($\pm$0.0) |
| $\eta = 0.01$ | Random | 0 | 0 | 0 | 0 | 0 | 3 | 0.0 ($\pm$0.0) | 0.0 ($\pm$0.0) | 0.6 ($\pm$0.5) |
| | AIT | 0 | 0 | 0 | 0 | 0 | 0 | 0.0 ($\pm$0.0) | 0.0 ($\pm$0.0) | 0.0 ($\pm$0.0) |
| $\eta = 0.02$ | Random | 0 | 4 | 0 | 0 | 0 | 12 | 0.0 ($\pm$0.0) | 0.0 ($\pm$0.0) | 6.0 ($\pm$1.6) |
| | AIT | 0 | 0 | 0 | 0 | 0 | 3 | 0.0 ($\pm$0.0) | 0.0 ($\pm$0.0) | 0.0 ($\pm$0.0) |
| $\eta = 0.05$ | Random | 1 | 9 | 0 | 2 | 1 | 33 | 1.3 ($\pm$0.5) | 8.0 ($\pm$2.2) | 27.0 ($\pm$0.5) |
| | AIT | 0 | 7 | 0 | 0 | 0 | 23 | 0.0 ($\pm$0.0) | 1.3 ($\pm$0.5) | 18.7 ($\pm$1.2) |
| $\eta = 0.1$ | Random | 9 | 9 | 9 | 16 | 16 | 45 | 11.0 ($\pm$0.8) | 20.7 ($\pm$0.5) | 40.0 ($\pm$0.8) |
| | AIT | 7 | 9 | 6 | 16 | 15 | 44 | 10.3 ($\pm$0.5) | 20.0 ($\pm$0.8) | 39.3 ($\pm$1.2) |

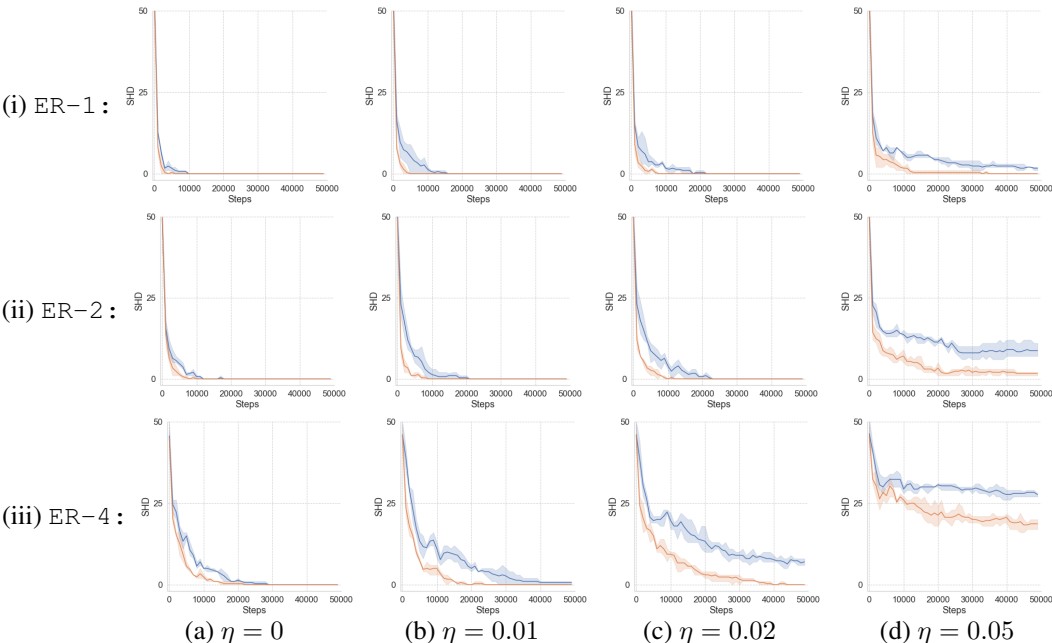

(i) ER-1:

(ii) ER-2:

(iii) ER-4:

(a) $\eta = 0$     (b) $\eta = 0.01$     (c) $\eta = 0.02$     (d) $\eta = 0.05$

Figure 17: Convergence behaviour in terms of SHD for random ER graphs of various densities over 10 variables under different noise levels $\eta$. Overall, Active Intervention Targeting (orange) clearly outperforms Random Targeting (blue) over all densities under all noise levels. The performance gap becomes of larger magnitude as density of the graph and the noise level increases. Error bands were estimated using 3 random ER graphs per setting.

### A.6.7 IDENTIFICATION OF INFORMATIVE INTERVENTION TARGETS

Our proposed method aims to select most *informative* intervention target(s) $I_{k*} \in I$ with respect to identifiability of the underlying structure. We conjecture that such targets yield relatively high discrepancy between samples drawn under different hypothesis graphs, indicating larger uncertainty about the target node's relation to its parents and/or children.

In order to evaluate our methods capability of detecting informative intervention targets, we perform multiple experiments on structured graph structures (`chain5`, `tree5` and `full5`) where we preinitalize the structural belief to the ground-truth structure structure but keeping one edge between a pair of nodes $(i,j)$ undirected, i.e. $\sigma(\gamma_{i,j}) = \sigma(\gamma_{j,i}) = 0.5$. Throughout the experiments, we vary the position of the undirected edge and analyze which nodes are targeted by our method.

Over all evaluated settings, we can observe how AIT preferentially targets the pair of nodes corresponding to the undirected edge, with small preferences towards the source nodes of the correct directed edge (see in Figure 18 and Figure 19). This observation is in line with our conjecture that AIT preferentially targets nodes with larger uncertainty about the target node's relation to its parents and/or children.

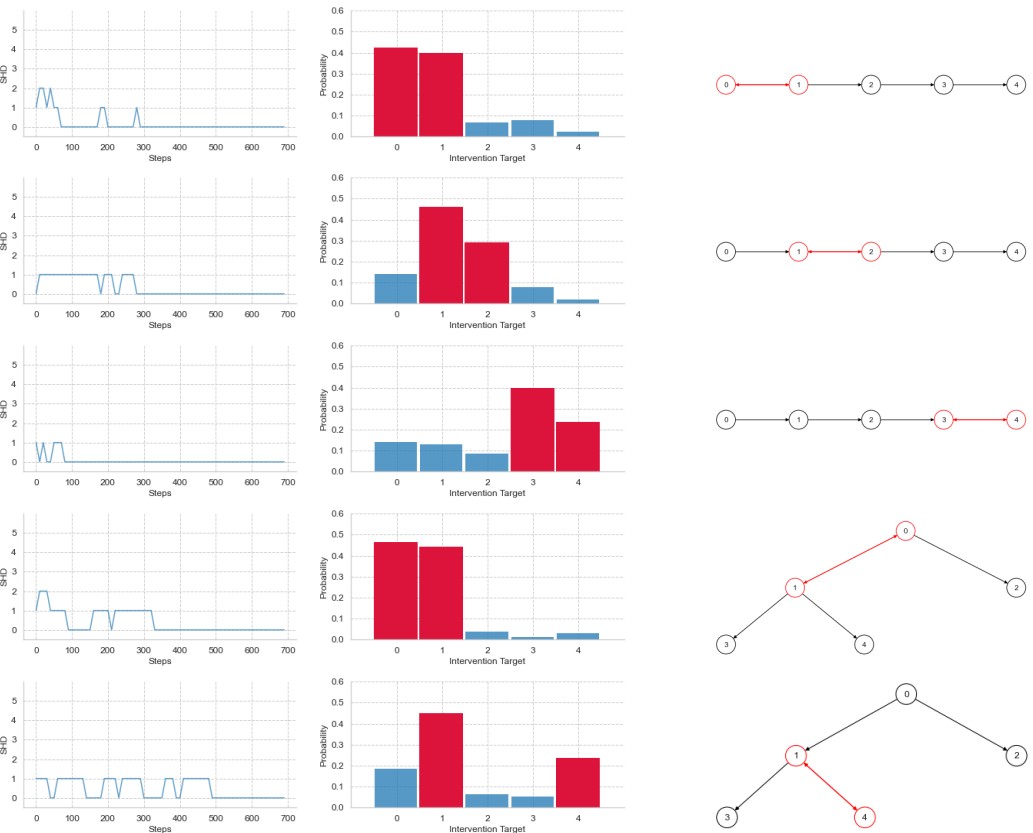

Figure 18: AIT chooses informative intervention targets by preferentially identifying and targeting the pair of nodes corresponding to the undirected edge (nodes are marked red in the distribution of selected target nodes and edges is visualized red in the graph on the right). - First set of experiments based on the structured graphs `chain5` and `tree5`.

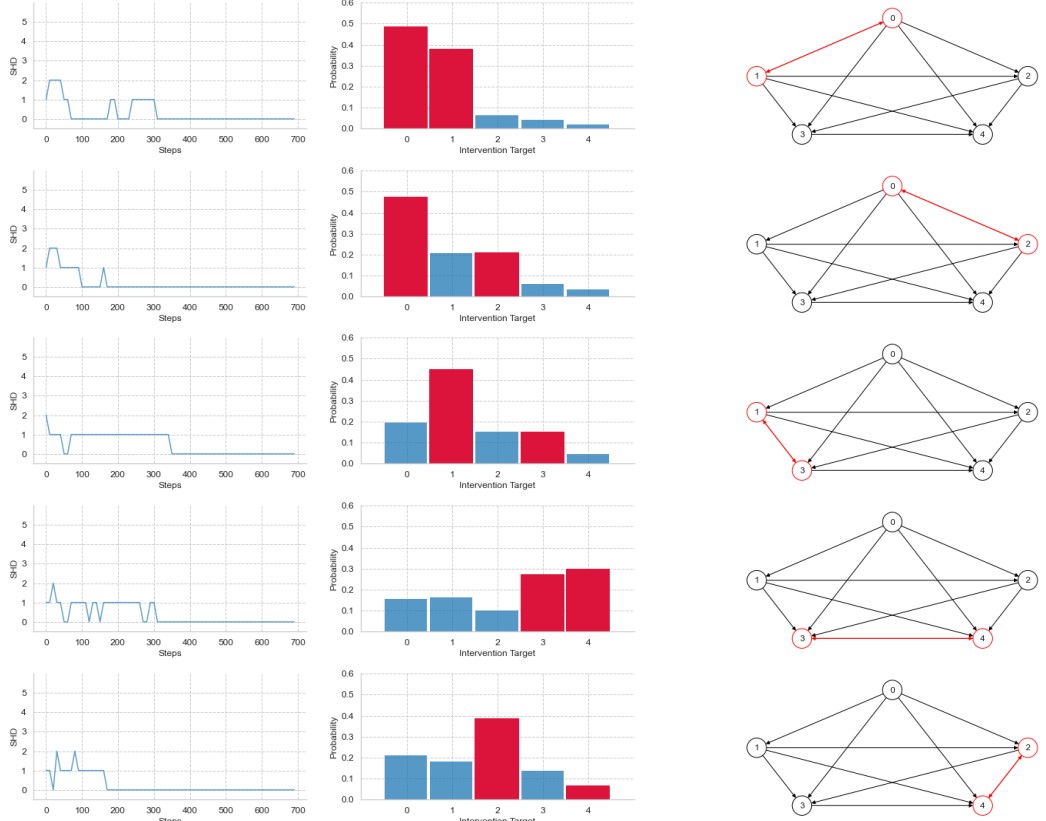

Figure 19: AIT chooses informative intervention targets by preferentially identifying and targeting the pair of nodes corresponding to the undirected edge (nodes are marked red in the distribution of selected target nodes and edges is visualized red in the graph on the right). - Second set of experiments based on the structured graph `full5`.

### A.6.8   LIMITED INTERVENTION TARGETS

While we allow access to all possible single-target interventions in all other experiments, real world settings are usually more restrictive. Specific interventions might be either technically impossible or even unethical, or the experiments might want to prevent interventions upon specific target nodes due to increased experiment costs. In order to test the capability of AIT, we limit the set of possible intervention targets in the following experiments and analyze the resulting behaviour based on DSDI. We examine speed of convergence and the effect on the target distribution under different scenarios on structured graphs using DSDI with AIT based on single-target interventions.

**Scenario 1:** We perform experiments on a `Chain5` graph where we restrict us on intervening upon a different node in five experiment and once allow access to all targets as a comparison.

Throughout the experiments, we observe that blocking interventions on nodes of a higher topological level results in greater degradation of the convergence speed compared to blocked intervention on lower levels (see Figure 20). Furthermore, the distribution of selected targets indicates that our method preferentially chooses neighboring nodes of a blocked target node in the restricted setting.

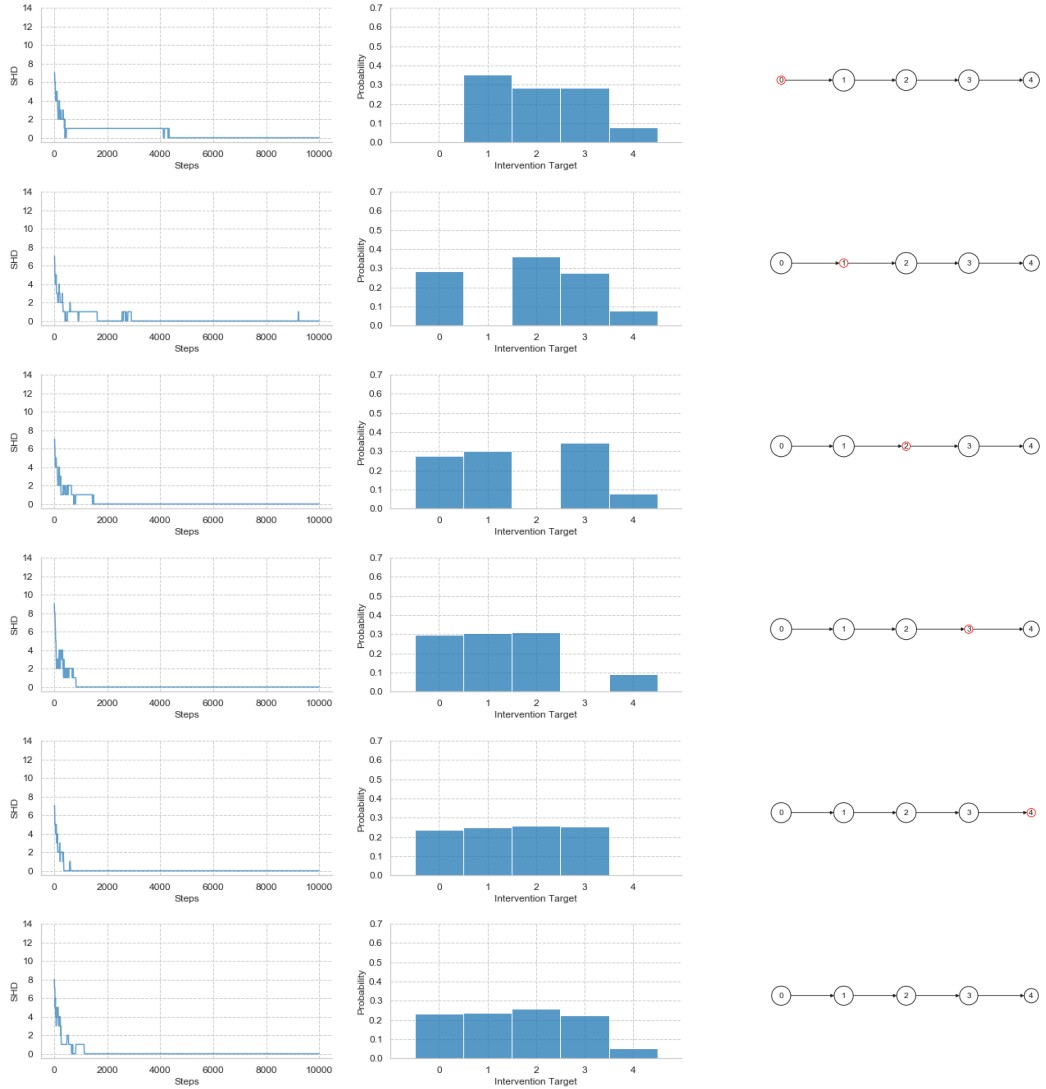

Figure 20: Limited intervention targets on `Chain5`: The impact of the restricted target node (red circled node) is clearly observable in the convergence speed (left) and distribution of target selections (middle). The speed of convergence indicates a dependence on the topological characteristic of the restricted intervention target.

**Scenario 2:** We perform multiple experiments on a `Tree5` graph where we restrict access to different subsets of nodes (e.g. root node, set of all sink nodes) for single-target interventions.

Similar to the experiments on `Chain5`, we observe a clear impact of the available intervention targets on the convergence speed and identifiability of the underlying structure (see Figure 21). While preventing interventions on all sink nodes (node 2, 3 and 4) results in improved convergence towards the underlying structure, restricted access to the set of nodes which act as causes of other nodes (node 0 and 1) prevents us from identifying the correct underlying structure.

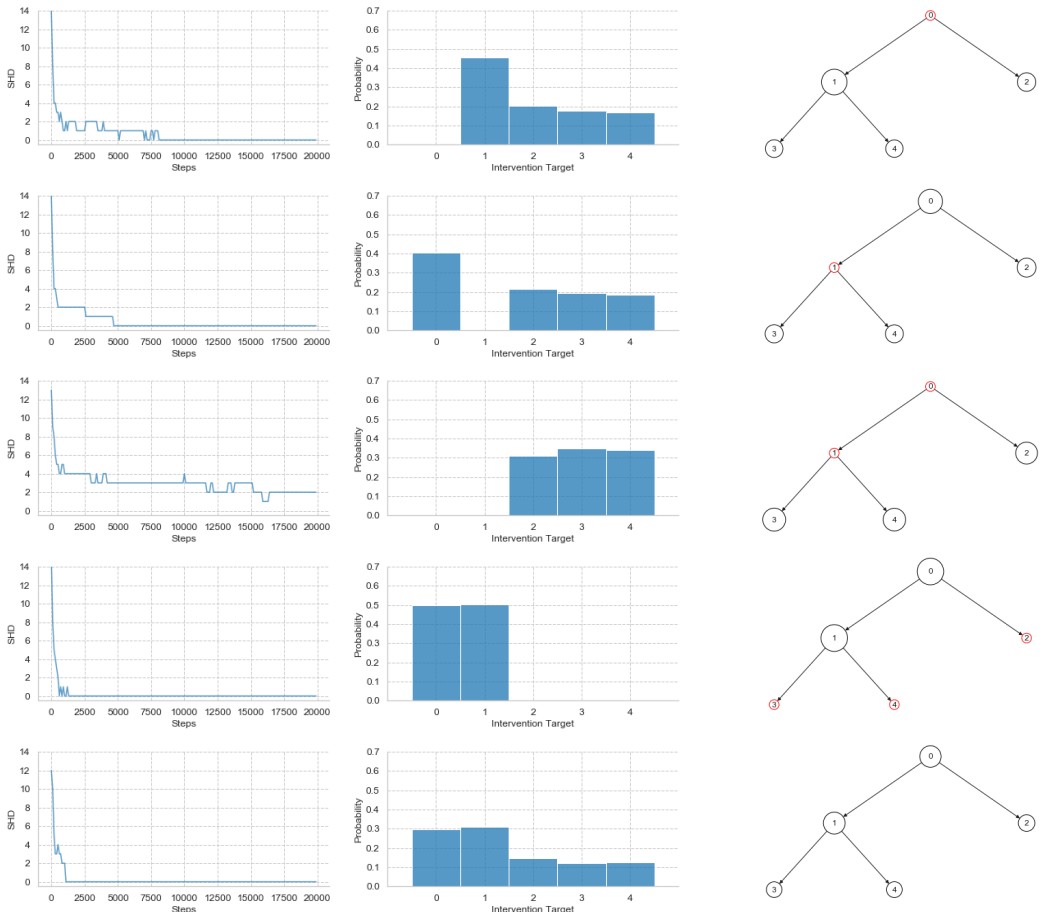

Figure 21: Limited intervention targets on `Tree5`: The impact of the restricted target nodes (red circled nodes) is clearly observable in the convergence speed (left) and distribution of target selections (middle).

## A.7 Continuous Setting: Technical Details and further Results

While the original framework of DCDI (Brouillard et al., 2020) proposes a joint-optimization over the observational and interventional sample space by selecting samples at random, we adapt their framework to the setting of active causal discovery where we acquire interventional sample in an adaptive manner. We hypothesize that a controlled selection of informative intervention targets allows a more rapid and controlled discovery of the underlying causal structure.

### A.7.1 Integration of AIT into DCDI

Instead of demanding the full interventional target space during the complete optimization as in the original approach, we split the optimization procedure into different episodes, where AIT is used to estimate a target space $I$ of size $K$ for each episode. This is done by computing the discrepancy scores over all possible intervention targets and selecting the $K$ highest scoring targets. During an episode, we continue by performing $L$ gradient steps using the fixed target space $I$ and reevaluate it afterwards for the next episode. We visualize the adaption in the following high-level outline of the individual methodologies.

**Algorithm 3** DCDI

1: $I \leftarrow$ Full target space of size $K = N$
2: Run DCDI on $I$ until convergence

$\Rightarrow$

**Algorithm 4** DCDI + AIT

1: **for** episode $e = 0$ **until** convergence **do**
2:     $I \leftarrow$ Estimate target space of size $K$ using AIT
3:     Run $L$ gradient steps of DCDI on $I$
4: **end for**

### A.7.2 Evaluation

We evaluate the effectiveness of AIT in the base framework of DCDI in the setting of non-linear, continuous data generated from random graphs over $N = 10$ variables and show the potential of our proposed method.

**Structural Identification / Convergence:** Despite their joint optimization formulation is not apriori designed for the setting of experimental design, an AIT guided version shows superior/competitive performance in terms of structural identification and sample complexity over the original formulation (see Figure 22).

**Distribution of Intervention Targets:** As in DSDI, we observe strong correlation of the number of target selections with the measured topological properties of the specific nodes. This indicates a controlled discovery of the underlying causal structure through preferential targeting of nodes with greater (downstream) impact on the overall system. In addition, interventions on variables without children are drastically reduced (see also §4 for equivalent observations in DSDI).

**Effect of Target Space Size** $K$**:** While the original formulation assumes $K = N$ for the complete optimization procedure (i.e. $L = 1$) and relies on random samples out of the full target space, our adapted AIT-guided version of DCDI constrains the target space to a subset of targets for each episode. An ablation study on the size of the target space shows that for all choices of $K \in \{2, 4, 6, 8\}$, our approach outperforms the original formulation in terms of sample complexity while achieving same or better performance in terms of SHD.

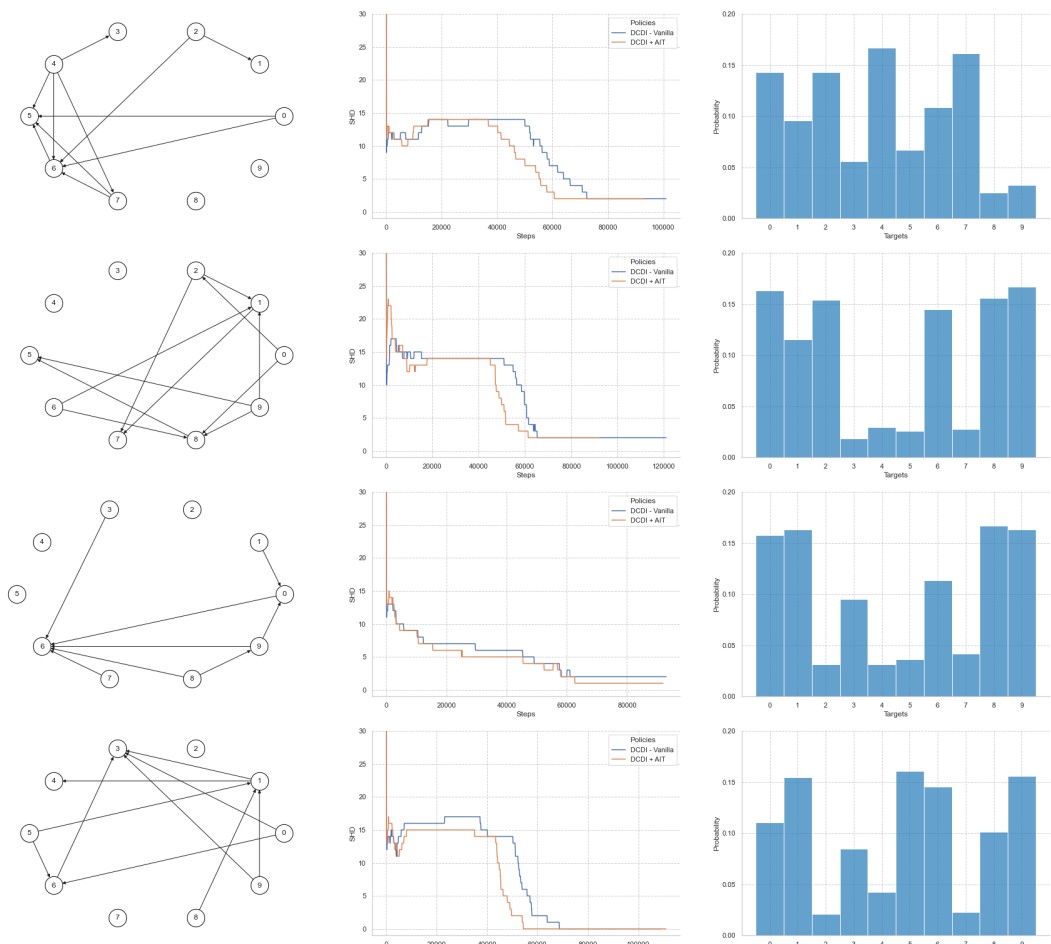

Figure 22: While DCDI Vanilla assumes access to the full interventional target space through the complete optimization, the AIT guided DCDI approach reevaluates its interventional target space of size $K = 6$ every $L = 1000$ gradient steps. Among the above evaluated graphs (ground-truth on the left), DCDI+AIT demonstrates a more rapid identification of the underlying causal structure while achieving same or better performance in terms of SHD. The distribution of selected single-node intervention targets reveals again its connection to the topological properties of the corresponding nodes.

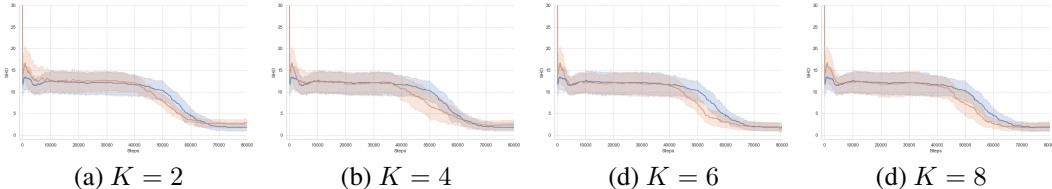

| (a) $K = 2$ | (b) $K = 4$ | (d) $K = 6$ | (d) $K = 8$ |

Figure 23: All evaluated target space sizes $K \in \{2, 4, 6, 8\}$ show that DCDI+AIT (orange) out-performs DCDI (blue) in terms of sample complexity while achieving same or better performance. Error bands were estimated using 10 random ER graphs per setting.

