# OpenReview forum: "Learning Neural Causal Models with Active Interventions"
_ICLR.cc/2022/Conference — ICLR 2022 Submitted_

### Official Review · Reviewer_cZsR · 2021-10-30

**Correctness:** 2
**Technical Novelty And Significance:** 2
**Empirical Novelty And Significance:** 2
**Recommendation:** 3
**Confidence:** 2

**Main Review:**

Sampling from interventional distributions and quantifying its difference with the available data is creative as an approach to guide the selection of interventional targets. Unfortunately, the paper is very hard to follow and seems to rely excessively on DSDI, both to understand the proposed approach and in applications despite the authors claiming that the proposal can be used with any differentiable causal discovery algorithm (contributions). The exact procedure is complex, it involves sampling DAGs and parameters, and should be clarified.

Am I correct in assuming that in every training iteration of the causal discovery algorithm, the proposed approach samples DAGs, interventional distributions, and samples from each interventional distribution to be compared using the proposed score? What is the computational complexity?

Along these lines, the exact procedure is unclear. Algorithm 1 vaguely mentions "perform an intervention" on line 3 without details of what kind of interventions are considered (e.g. do, soft, what exact values are set, etc.). Next, it is meant to draw samples from $\mathcal P$ which is defined as a set of functional parameters which I understand are conditional distributions. How are the conditional distributions chosen? What is the functional relationship between parents and children?

My main concern is about the claim that one can meaningfully sample from constructed interventional distributions. For this to be valid one would need access to the underlying structural causal model in general, which is far beyond the DAG we seek to estimate in the first place - I don't believe that the constructed interventional distributions give any useful guidance to select interventions as interventional distributions are not identified in general. The use of the term "interventional data" I find misleading as nowhere in the algorithm description of experiments (unless I missed it) do the authors make use of a data source in which an experiment has been performed.

One argument repeatedly made in the paper is that existing structure learning algorithms are computationally expensive. Yet, experimental evaluations are made on graphs of less than 20 nodes, on relatively simple graphs, and without run time comparisons. This claim and others are far from justified.

If the main contribution is the score, this is not sufficiently discussed or justified. What makes this a good score? Do we have any guarantees that it favors good interventions?


**Summary Of The Paper:**

Structure learning from observational data is a long-standing challenge that could benefit from creative uses of interventional data. The authors explore the use of active learning in a continuous optimization framework to better traverse and narrow down the space of potential graphs that describe the data. The contribution is a heuristic to choose the "best" intervention target based on a comparison between samples from an intervened SCM with the current best guess of an underlying causal discovery algorithm (in which the proposal is embedded).

**Summary Of The Review:**

Ultimately my opinion is that without additional theoretical results or better justification and exposition of the proposed approach to advance the state of the art, the contribution is limited.

---

> ### Author Response · Authors · 2021-11-20
> **Author Response [1/3]: Official Review of Paper1400 by Reviewer cZsR**
>
> We would like to thank the reviewer for their feedback and appreciate the interest in our proposed approach. We find it unfortunate to hear that our paper is difficult to follow. Thus, we would like to take the opportunity to clarify as many questions as possible with the reviewer in order to improve our paper.
>
> In particular we would be happy about further comments on what renders the paper difficult to read. Would an extended review of the base frameworks (DSDI and DCDI) improve the understanding?
>
> > “Unfortunately, the paper is very hard to follow and seems to rely excessively on DSDI, both to understand the proposed approach and in applications despite the authors claiming that the proposal can be used with any differentiable causal discovery algorithm (contributions). The exact procedure is complex, it involves sampling DAGs and parameters, and should be clarified.”
>
> The methodology of the current available causal discovery framework from fused data (DSDI, DCDI and ENCO) is quite extensive and does not allow a fully comprehensive review within the main text of our work. While we currently only summarize the individual methodologies in a compact way in the main text, we intend to provide an extended review of the methods in the appendix in the final version of our manuscript. In addition, we illustrated the integration / applicability of our method in the above figure.
>
> > “Am I correct in assuming that in every training iteration of the causal discovery algorithm, the proposed approach samples DAGs, interventional distributions, and samples from each interventional distribution to be compared using the proposed score?”
>
> In the setting of DSDI, we currently evaluate the discrepancy scores of available intervention targets in every iteration of structural fitting. This involves:
>  - Sampling a set of hypothesis DAGs
>  - For every considerate intervention:
>      - Sampling a fixed number of samples on every hypothesis graph .
>
> Note that we do not have to sample interventional distributions. Based on the topological order of the hypothesis DAGs under a certain intervention, we are able to generate samples from an interventional distribution by using ancestral sampling from the learned conditionals (each conditional is represented by a MLP where we mask off all non-parent nodes in the input of a specific variable). In the discrete setting, the variables of the intervened nodes are sampled from an uniform distribution over all possible assignments, and from N(2,1) in the continuous setting.
>
> The adapted version of DCDI which integrates AIT reevaluates the discrepancy scores of available intervention targets every episode. An episode length is defined as a hyperparameter and can be varied flexibly. The frequency of the discrepancy score evaluation is in general a design choice and can be flexibly adjusted.
>
>
> > “What is the computational complexity?”
>
> We will update the submitted manuscript with a computational complexity table. An execution of AIT in a setting with N variables requires following steps:
>
> - Sampling hypothesis DAGs
>
>   → O( $NB_{Graphs}$ * N^2 )
>
> - Compute topological order for all hypothesis DAGs
>
>    → O( $NB_{Graphs}$  * N^2 )
>
> - Sampling interventional samples for all considered interventions on all hypothesis graphs
>
>   →O( $NB_{Graphs}$  * $NB_{ConsideredInterventions}$  * N *  $FP_{FuncFit}$(BS = $NB_{SamplesPerIntervention}$))​
>
> - Score Computation
>
>   →O(  $NB_{Graphs}$ * $NB_{ConsideredInterventions}$  *  $NB_{SamplesPerIntervention}$ * N^2 )
>
> where:
> - $BS$ := Batch Size
> - $FP_{FuncFit}$ := Foward pass of Functional Fitting per variable (e.g. one forward pass of 2 Layer MLP)
> - $NB_{Graphs}$  := Number of Graphs
> - $NB_{ConsideredInterventions}$  := Number of considered Interventions that need to to be evaluated
> - $NB_{SamplesPerIntervention}$ := Number of samples to be generated from each interventional setting

---

> > ### Author Response · Authors · 2021-11-20
> > **Author Response [2/3]: Official Review of Paper1400 by Reviewer cZsR**
> >
> > > “Along these lines, the exact procedure is unclear. Algorithm 1 vaguely mentions "perform an intervention" on line 3 without details of what kind of interventions are considered (e.g. do, soft, what exact values are set, etc.).“
> >
> > We thank the reviewer for pointing out this missing point in our experimental setup. We will update the manuscript.
> > Our approach is agnostic to the choice of the interventions. In the current set of experiments we rely on random interventions and perform the same type of interventions as we perform on the ground truth structures. In the discrete case, we sample the intervened variable from a uniform distribution over all assignments. In the continuous setting, we sample the interventions as N(2,1).
> >
> > > “Next, it is meant to draw samples from P which is defined as a set of functional parameters which I understand are conditional distributions. How are the conditional distributions chosen? What is the functional relationship between parents and children?”
> >
> > The conditional distributions are represented through MLPs (one MLP per conditional). For example in the setting of $p(X_j | pa(X_j))$, we mask all non-parent nodes of variable j to 0 in the input to the MLP. Therefore we guarantee that only the values of parent nodes are used for the prediction of $X_j$'s value.
> > In order to draw interventional samples, we execute following procedure:
> >
> > - For every considered intervention target I:
> >   - For every hypothesis DAG G:
> >     - G_inter ← Adjust G according to the chosen intervention target (delete incoming edges to represent the perfect intervention setting)
> >     - Ancestral Sampling according topological order of G_inter (this step depends on the chosen base framework)
> >
> > Example: Ancestral Sampling in DSDI:
> >
> > - For node n in [topologically ordered nodes]:
> >   - If n == intervened node:
> >     - Sample values of intervened node (from interventional distribution)
> >   - Else:
> >      - Sample values for node n through a forward pass of the corresponding MLP and masking off non-parent nodes of node n (according to the intervened hypothesis graph G_inter)
> >
> >
> > > “My main concern is about the claim that one can meaningfully sample from constructed interventional distributions. For this to be valid one would need access to the underlying structural causal model in general, which is far beyond the DAG we seek to estimate in the first place - I don't believe that the constructed interventional distributions give any useful guidance to select interventions as interventional distributions are not identified in general. “
> >
> > We do not exactly understand what the reviewer means with this concern and politely ask for clarification on this point.
> >
> > Our sampling approach is based on conditionals which are fitted to observational data. Given a specific intervention on variable $X_j$, we generate the corresponding interventional samples by ancestral sampling under different hypothesis graphs. In the discrete setting, the variables of the intervened nodes are sampled from an uniform distribution over all possible assignments, and from N(2,1) in the continuous setting. All other variables are computed using a forward pass of the MLPs which represent $p(X_i | pa(X_i))$.
> >
> > If samples from an intervention on a certain node exhibit large variance across different hypothesis graphs in comparison to other interventions, it indicates that we are unsettled about the relation of this corresponding node to its children. If we would be settled about the relations to its children, the variance among different graphs would be smaller as in the unsettled case. We will provide more intuition using visualizations in an update of the manuscript for further clarification.
> >
> > > “The use of the term "interventional data" I find misleading as nowhere in the algorithm description of experiments (unless I missed it) do the authors make use of a data source in which an experiment has been performed.”
> >
> > As indicated earlier, we find it unfortunate to hear that our paper is difficult to follow. This and further comments hints that the reviewer did not fully understand our method due to a missing understanding of the base frameworks (DSDI and DCDI). We refer to the compact description of DSDI and DCDI to highlight where interventional data is used. For improved clarity, we intend to provide a more insightful visualization of our proposed approach alongside an extended review of the base frameworks in the appendix in the final version of our manuscript. We expect that a more extensive coverage of the methodologies of the base framework will clarify most of the reviewer’s concerns.

---

> > > ### Author Response · Authors · 2021-11-20
> > > **Author Response [3/3]: Official Review of Paper1400 by Reviewer cZsR**
> > >
> > > > “One argument repeatedly made in the paper is that existing structure learning algorithms are computationally expensive. Yet, experimental evaluations are made on graphs of less than 20 nodes, on relatively simple graphs, and without run time comparisons. This claim and others are far from justified.”
> > >
> > > We thank the reviewer for this comment and agree that this statement might be misleading. It is true that we may not be as computationally efficient as every non-neural network approach, but we primarily address sampling efficiency in our work and not necessarily computational efficiency. We will correct it in an update of our manuscript.
> > >
> > > Our active learning algorithm operates on neural causal models (with interventional data). We demonstrate how a targeted discovery improves sample-efficiency of existing neural causal discovery approaches. Also, note that though neural network approaches are not as computationally efficient as non-neural network approaches, they do converge to significantly better results in our evaluated setting of non-linear data.
> > >
> > > > “If the main contribution is the score, this is not sufficiently discussed or justified. What makes this a good score? Do we have any guarantees that it favors good interventions?”
> > >
> > > We acknowledge that we have not provided theoretical justifications or guarantees for our method. We believe that missing theory in a machine learning work should not prevent a paper from its acceptance and all sources of evidence should be taken into consideration. In addition, we are confident that the wide applicability of our approach and the strong empirical results (in noise-free and noise-perturbed settings) serves as a novel and interesting research result on its own. We would be happy to investigate the theoretical frameworks as a future work

---

> > > > ### Comment · Reviewer_cZsR · 2021-11-25
> > > > **Thank you for all clarifications**
> > > >
> > > > Thank you for taking the time to address in detail all of my questions. It appears that I am misunderstanding some of the details in the interaction between making an intervention, not having the underlying SCM, and still improving upon structure learning with observational data only. Perhaps a more comprehensive description of the approach this paper relies on, as the authors suggest, would be helpful. I will decrease my confidence score to reflect this fact, even though I maintain the opinion that the paper lacks sufficient theoretical justification (especially important in unsupervised problems) and that the paper would benefit from an additional round of revisions to improve the quality of the exposition.

---

### Official Review · Reviewer_KxZA · 2021-10-30

**Correctness:** 3
**Technical Novelty And Significance:** 3
**Empirical Novelty And Significance:** 2
**Recommendation:** 5
**Confidence:** 4

**Main Review:**

Overall, the proposed method is interesting and novel. However, since the proposed method is based on a heuristic without theoretical analysis, as also acknowledged by the authors, I think it is important to demonstrate empirically with a fair experiment setup that the proposed method works well compared to the existing baselines. I am willing to increase my score if the authors are able to address my comments below.

Strengths:
- The paper is well written.
- The proposed method is interesting and novel.

Weaknesses:
- The proposed method lacks theoretical analysis.
- It is not surprising that GES, NOTEARS, and DAG-GNN do not perform well, as they are based on observational data. I would encourage the authors to include more baselines that handle interventional data, such as IGSP [1] or UT-IGSP [2], together with nonparametric test.
- It would be better to provide what score is used for GIES and GES. Have the authors considered the generalized score [3] that better handles nonparametric relationship for a fairer comparison?
- Based on my understanding, the authors used the original ICP [4] that is based on linear model. It may be better to also include its nonlinear variant [5].
- Figure 23 seems to show that DCDI performs better than DCDI+AIT. It would be better to include similar experiments in Tables 1 and 2 for DCDI.
- How did the authors ensure a fair comparison between the proposed method based on active intervention, and the baselines based on observational data and/or random interventions? E.g., what are the number of samples/interventions used for both cases?
- The paper would be stronger if it includes the structural intervention distance (SID) [6] that may also be informative.

Minor comments:
- Based on my understanding, the experiments in Tables 1 and 2 are based on discrete data. It would be better to explain how NOTEARS handles discrete data.
- For the real datasets considered in Section 4, did the authors experiment with the original datasets, or only adopt their ground truth causal graphs and generate synthetic data based on these graphs? It would be better to make this clear in the main paper. The current description sounds like it is the former (e.g., the last two sentences in the subsection "Structure discovery: flow cytometry and asia dataset"), although it seems unclear how active interventions could be applied there.

References:
1.  Permutation-based Causal Inference Algorithms with Interventions, 2017.
2.  Permutation-Based Causal Structure Learning with Unknown Intervention Targets, 2020.
3. Generalized Score Functions for Causal Discovery, 2018.
4.  Causal inference using invariant prediction: identification and confidence intervals, 2015.
5.  Invariant Causal Prediction for Nonlinear Models, 2018.
6.  Structural Intervention Distance (SID) for Evaluating Causal Graphs, 2014.

-----------
After reading the authors' response and the other reviews, my concerns about the experiment setup persist. I agree with Reviewer ma3K that some part of the experiments could be further improved to ensure a fair experiment setup.


**Summary Of The Paper:**

The paper proposes a method to select interventions that enable efficient identification of the underlying causal structure. In particular, the method picks the intervention that exhibits the highest discrepancies between post-interventional sample distributions generated. The authors provide experiment results to demonstrate that the proposed method has a better performance over the other baselines, and also improves the sample efficiency.

**Summary Of The Review:**

Overall, the proposed method is interesting and novel. However, since the proposed method is based on a heuristic without theoretical analysis, as also acknowledged by the authors, I think it is important to demonstrate empirically with a fair experiment setup that the proposed method works well compared to the existing baselines.

---

> ### Author Response · Authors · 2021-11-16
> **Response: Official Review of Paper1400 by Reviewer KxZA**
>
> We would like to thank the reviewer for their thorough and valuable feedback which will help us considerably in improving our paper. We are grateful for the interest in our proposed method and appreciate their support for the writing and the novelness of the proposed approach.
>
> We would like to address the reviewer’s concerns as follows:
>
> > “The proposed method lacks theoretical analysis.”
>
> As indicated by the reviewer, we acknowledge that we have not provided theoretical justifications for our method. We believe that missing theory in a machine learning work should not prevent a paper from its acceptance and all sources of evidence should be taken into consideration. In addition, we are confident that the wide applicability of our approach and the strong empirical results (in noise-free and noise-perturbed settings) serves as a novel and interesting research result on its own. We are happy about the reviewer's pointers to further improve our empirical evaluation. Moreover, we would also be happy to investigate the theoretical frameworks as a future work.
>
> > “It is not surprising that GES, NOTEARS, and DAG-GNN do not perform well, as they are based on observational data. I would encourage the authors to include more baselines that handle interventional data, such as IGSP [1] or UT-IGSP [2], together with nonparametric test.”
>
> We thank the reviewer for pointing us to the work of IGSP or UT-IGSP. We will include the two methods in our comparison study in the final version of the manuscript.
>
> We understand the point that the comparison with methods that only handle observational data is not fully fair. As our proposed method is embedded in continuous optimization approaches, we intended to compare its performance against existing differentiable approaches  nd included the work of NOTEARS and DAG-GNN. However, up to the current time, not a lot of differentiable methods support the setting of observational and interventional data.
>
> > “It would be better to provide what score is used for GIES and GES. Have the authors considered the generalized score [3] that better handles nonparametric relationship for a fairer comparison?”
>
> We thank the reviewer for pointing us to the generalized score. We used the “GaussL0penIntScore” score in GIES for the existing experiments. We will rerun the experiments with the generalized score and update the results if favourable results are achieved over the “GaussL0penIntScore” score. We will provide an extended section about the used baselines in the final manuscript and will highlight technical details and possible issues.
>
> > “Based on my understanding, the authors used the original ICP [4] that is based on linear model. It may be better to also include its nonlinear variant [5].”
>
> We are using the original ICP version for comparison. We do not show any comparison with the non-linear ICP version, as the linear ICP outperformed its non-linear version in all settings in the original work of DSDI.
>
> > “Figure 23 seems to show that DCDI performs better than DCDI+AIT. It would be better to include similar experiments in Tables 1 and 2 for DCDI.”
>
> Thanks for pointing this out. We realize that the description of Figure 23 might be a bit misleading. While the blue curve (DCDI) always relies on the full intervention spectrum, we test the adapted version of DCDI within an adaptive experimental design setting with multiple target space sizes (= number of single-target interventions per episode). While DCDI (vanilla) is formalized as a joint optimization formulation over the full target space, we show that our extension already matches the performance of DCDI with a target space of size K=6, which is considerably smaller than the full target space of K=10. Thus, we show DCDI’s possible extension to experimental design based on our method. However, we are happy to add a comparison table for improved clarity.
>
> >  “How did the authors ensure a fair comparison between the proposed method based on active intervention, and the baselines based on observational data and/or random interventions? E.g., what are the number of samples/interventions used for both cases?”
>
> We have limited ourselves to following setting: 100’000 Observational samples and 1000 samples per interventional setting
>
>
> Overall, we hope that we could address all your questions. Please let us know if there are any other concerns of yours which we can address.

---

### Official Review · Reviewer_ma3K · 2021-11-01

**Correctness:** 3
**Technical Novelty And Significance:** 2
**Empirical Novelty And Significance:** 2
**Recommendation:** 5
**Confidence:** 4

**Main Review:**

Strengths:
1. The paper is written clearly;
2. the tackled problem is well-motivated;
3. I think the idea that select the interventional variable by maximizing the disagreement between the post-interventional sample distributions under the hypothesis graphs is novel.

Weaknesses:
One main aspect where some improvements could make the paper better is the experiment part. The comparison results between the proposed method and DCDI are supportive in only verifying that the active strategy is better than random selection. However, according to my understanding, although the novel point of this paper is to present an active causal discovery method based on continuous optimization framework, the tackled problem is still actively causal discovery. In the literature, there have been many classical methods regarding actively causal discovery (e.g. Active learning of causal networks with intervention experiments and optimal designs, Two optimal strategies for active learning of causal models from interventional data). Hence, in addition to illustrating that under continuous optimization framework the active strategy works, it is necessary to illustrate that the proposed method could preform better than the previous active causal discovery methods, which convince the readers that the proposed method is the best in this kind of tasks. In addition, could the authors give more details about the baselines? The baselines do not quite match the tasks in this paper, hence I want to know some execution details. For example, GIES is for obser. + passive inter., the active strategy is not involved. How do the authors select the intervention variable? ICP does not exploit the information about which variables are softly intervened (taken by dynamic environment). The comparison between the proposed method and ICP seems not quite fair. Do authors make some modifications? Towards active causal discovery, I think the two methods I mentioned above seem to be more suitable as baseline methods.

I have one another question. The authors claim that the proposed method is applicable for multi-node intervention. I agree with that. However, in this case the complexity of Line 2 of Algorithm 1 is quite large. How do the authors address this problem?

Other suggestions or typos:
1. I think it will be better if the authors give more details about training functional parameters in the paper or appendix. The current version made me have to read paper DSDI.

2. A missing full point in the paragraph of Assumptions (Page 3).

**Summary Of The Paper:**

In this paper, the authors take a further step by introducing active interventions in differentiable neural network-based methods. The intervention target is selected based on maximizing the disagreement between the post-interventional sample distributions under the hypothesis graphs. The experimental results verify the effectiveness of the active strategy compared to random selection.

**Summary Of The Review:**

The main reason that I give a slightly negative score in the current is that I think the experiments are not supportive enough. Different from some theoretical causality paper, this paper tackles a practical and widely researched problem. Hence I think the experiments need to be designed carefully. I look forward to more clues from the authors. If they could address my concerns and questions, I am very happy to increase my score.

---

> ### Author Response · Authors · 2021-11-16
> **Response: Official Review of Paper1400 by Reviewer ma3K**
>
> We would like to thank the reviewer for their thorough and valuable feedback which will help us considerably in improving our paper. We are grateful for the interest in our proposed method and appreciate their support for the writing and the novelness of the proposed approach.
>
> We would like to address the reviewer’s concerns as follows:
>
> > “One main aspect where some improvements could make the paper better is the experiment part. The comparison results between the proposed method and DCDI are supportive in only verifying that the active strategy is better than random selection. However, according to my understanding, although the novel point of this paper is to present an active causal discovery method based on continuous optimization framework, the tackled problem is still actively causal discovery. In the literature, there have been many classical methods regarding actively causal discovery (e.g. Active learning of causal networks with intervention experiments and optimal designs, Two optimal strategies for active learning of causal models from interventional data). Hence, in addition to illustrating that under continuous optimization framework the active strategy works, it is necessary to illustrate that the proposed method could preform better than the previous active causal discovery methods, which convince the readers that the proposed method is the best in this kind of tasks.”
>
> We agree with the reviewer that it would be interesting to compare our proposed methodology with existing work in experimental design. In order to provide a fair comparison of our approach to non-continuous frameworks and draw insightful conclusions, it would demand certain design changes (e.g. fixing an underlying skeleton, changing data generative mechanism (linear vs. non-linear data), assumption of infinite interventional samples, ... ) to match the underlying assumptions / requirements of each framework. We intend to add additional experiments to the final manuscript (e.g. by assuming access to the underlying ground-truth skeleton and compare against existing graph-theoretical approaches). Nevertheless, we would like to point out that our method is actually designed to work without this assumption. Therefore, we believe that a systematic comparison / benchmarking of different experimental design methods (non-differentiable and differentiable) should be covered in a work on its own since it would significantly increase the scope of our work.
>
> > “In addition, could the authors give more details about the baselines? The baselines do not quite match the tasks in this paper, hence I want to know some execution details. For example, GIES is for obser. + passive inter., the active strategy is not involved. How do the authors select the intervention variable?”
> We like to apologize for missing to add further experimental details and we will update this in the final version of the manuscript.
>
> We run GIES on observational and interventional data without any active strategy. We provide access to all possible single-target interventions (full single-target space) and limit the sample size to 1000 samples per interventional setting.
>
> >“ICP does not exploit the information about which variables are softly intervened (taken by dynamic environment). The comparison between the proposed method and ICP seems not quite fair.
>
> We understand the reviewers point and agree that our setting does not exactly match the setting of ICP. However, ICP is a common baseline in the setting of causal discovery and we provide the results just as comparison. We will provide an extended section about the used baselines in the final manuscript and will highlight possible issues.
>
> >“Do authors make some modifications to ICP?”
>
> We run ICP and Active-ICP on the same setting as GIES.
>
> >“I have one another question. The authors claim that the proposed method is applicable for multi-node intervention. I agree with that. However, in this case the complexity of Line 2 of Algorithm 1 is quite large. How do the authors address this problem?”
>
> We fully agree that the search space would be quite large in the setting of multi-target interventions. While we highlight our methods' applicability to such settings, we leave an efficient search in the target space as future work since it would significantly increase the scope of our submission.
>
> >“I think it will be better if the authors give more details about training functional parameters in the paper or appendix. The current version made me have to read paper DSDI.”
>
> We thank the reviewer for pointing this out. We will add an extended review of DSDI to the appendix in the final version of the manuscript.
>
>
> Overall, we hope that we could address all your questions. Please let us know if there are any other concerns of yours which we can address.

---

> > ### Comment · Reviewer_ma3K · 2021-11-17
> > **Response to the rebuttal**
> >
> > Dear authors:
> >
> >   Thank you very much for your response. However, I cannot quite agree with the opinions that the previous methods need the assumptions you mentioned.
> >
> >   About "fixing an underlying skeleton'': I think the related works do not assume that there is background knowledge about the skeleton. In the method of He & Geng, they learn a CPDAG with observational data before active intervention. And the method of "Two optimal strategies for active learning of causal models from interventional data'' additionally exploits the information of interventional data in estimating the skeleton.
> >
> >   To get the result about consistency, they possibly need the assumptions such as infinite interventional samples and observational samples. However, since there are no theoretical results in this paper, it is unclear whether such assumptions are needed in this paper if you want to conclude a similar result about consistency.
> >
> >   Hence, I still insist on the comments that the comparisons between the proposed method and the traditional graph-theoretical methods are necessary. And some details of experiments in the current version are lacked. I believe the paper will be better if the authors provide more details and convincing experiments results.
> >
> > Best Regards

---

### Decision · Program_Chairs · 2022-01-20

**Decision:**

Reject

**Comment:**

This paper proposes an active intervention-targeting mechanism for causal structure discovery. After the discussion, there was a consensus among the reviewers that this paper needs another round of revision to address the lingering concerns. These concerns include providing a more fair experimental setup (e.g. by properly distinguishing and designing proper experiments for the observational, random intervention, and targeted intervention settings). Since the paper lacks theoretical guarantees (which is OK and not a requirement for acceptance), the merits rest on providing a thorough and fair experimental evaluation.